**Multiple pathways for the formation of secondary organic aerosol in North China Plain**
**in summer**
Yifang Gu[1,3], Ru-Jin Huang[1,2,3], Jing Duan[1], Wei Xu[1], Chunshui Lin[1], Haobin Zhong[1,3], Ying
Wang[1], Haiyan Ni[1], Quan Liu[4], Ruiguang Xu[5,6], Litao Wang[5,6], Yong Jie Li[7]
[1]SKLLQG, Center for Excellence in Quaternary Science and Global Change, Institute of Earth
Environment, Chinese Academy of Sciences, Xi'an 710061, China
[2]Institute of Global Environmental Change, Xi'an Jiaotong University, Xi'an 710049, China
[3]University of Chinese Academy of Sciences, Beijing 100049, China
[4]State Key Laboratory of Severe Weather & Key Laboratory of Atmospheric Chemistry of C
MA, Chinese Academy of Meteorological Sciences, Beijing 100081, China
[5]Department of Environmental Engineering, School of Energy and Environmental Engineering,
Hebei University of Engineering, Handan 056038, China
[6]Hebei Key Laboratory of Air Pollution Cause and Impact, Handan 056038, China
[7]Department of Civil and Environmental Engineering, and Centre for Regional Oceans, Faculty
of Science and Technology, University of Macau, Taipa, Macau 999078, China
*Correspondence to*: Ru-Jin Huang (rujin.huang@ieecas.cn)
**Abstract**
Secondary organic aerosol (SOA) has been identified as a major contributor to fine
particulate matter ($PM_{2.5}$) in North China Plain (NCP). However, the chemical mechanisms
involved are still unclear due to incomplete understanding of its multiple formation processes.
Here we report field observations in summer in Handan of NCP, based on high-resolution
online measurements. Our results reveal the formation of SOA via photochemistry and two
types of aqueous-phase chemistry, the latter of which include nocturnal and daytime processing.
The photochemical pathway is the most important under high $O_x$ (=$O_3$ + $NO_2$) conditions (65.1
$\pm$ 20.4 ppb). The efficient SOA formation from photochemistry ($O_x$-initiated-SOA) dominated
the daytime (65% to OA) with an average growth rate of $0.8\,\mu g\,m^{-3}\,h^{-1}$. During the high relative
humidity (RH: 83.7 $\pm$ 12.5 %) period, strong nocturnal aqueous-phase SOA formation (aq-SOA)
played a significant role in SOA production (45% to OA) with a nighttime growth rate of 0.6
$\mu g\,m^{-3}\,h^{-1}$. Meanwhile, an equally fast growth rate of $0.6\,\mu g\,m^{-3}\,h^{-1}$ of $O_x$-initiated-SOA from
daytime aqueous-phase photochemistry was also observed, which contributed 39% to OA,
showing that photochemistry in the aqueous phase is also a non-negligible pathway in summer.
The primary-related-SOA (SOA attributed to primary particulate organics) and aq-SOA are
related to residential coal combustion activities, supported by distinct fragments from
polycyclic aromatic hydrocarbons (PAHs). Moreover, the conversion and rapidly oxidation of
primary-related-SOA to aq-SOA could be possible in the aqueous phase under high-RH
conditions. This work sheds light on the multiple formation pathways of SOA in ambient air of
complex pollution, and improves our understanding of ambient SOA formation and aging in
summer with high oxidation capacity.

KEYWORDS: secondary organic aerosol, aqueous-phase chemistry, photochemistry, multiple-
phase chemistry, complex air pollution

## 1. Introduction

Rapid economic growth and urbanization processes have led to severe particulate air pollution in China, affecting air quality, climates and human health (Huang et al., 2014; Cohen et al., 2017; An et al., 2019). Organic aerosol (OA) is a major component of aerosol particles, consisting of 20-90% of fine particle mass (Jimenez et al., 2009; Zhang et al., 2011). OA is either emitted directly from primary sources (referred to as primary OA, POA) such as traffic, cooking, coal combustion, and biomass burning, or produced through gas-to-particle conversion (referred to as secondary OA, SOA) in the atmosphere. In recent years, with the implementation of control measures, the POA fraction is decreasing and SOA fraction is increasing in North China Plain (NCP), indicating that SOA is becoming more critical for urban air quality (Huang et al., 2019; Xu et al., 2019; Gu et al., 2020). However, our understanding of the formation mechanisms and evolution processes of SOA is still limited.

Generally, SOA can be formed through gas-phase photochemical oxidation of volatile organic compounds (VOCs) followed by nucleation or condensation of oxidation products onto the preexisting particles (Donahue et al., 2006). Herndon et al., (2008) showed that oxygenated organic aerosol (OOA), a surrogate of SOA, was well correlated with odd oxygen ($O_x = O_3 +$ nitrogen dioxide ($NO_2$)) during photochemical processing. SOA can also be formed in the aqueous phase on wet aerosols, clouds and fogs through further chemical processes of water-soluble organic compounds or organic products of gas-phase photochemistry (Ervens et al., 2011, 2014). A growing number of laboratory studies and field measurements have indicated that aqueous-phase processes contribute efficiently to the formation of SOA (Gilardoni et al., 2016; Bikkina et al., 2017). However, how photochemistry and aqueous-phase chemistry coordinate to affect the formation of SOA is still unclear, despite numerous measurements to explore this question using aerosol chemical speciation monitor (ACSM) or aerosol mass spectrometer (AMS) (Hu et al., 2016b; Hu et al., 2017; Sun et al., 2016; Li et al., 2017; Sun et al., 2018b; Huang et al., 2019; Gu et al. 2020; Kuang et al., 2020). Field measurements in Beijing suggested that gas-phase photochemical oxidation can play a dominant role in SOA formation (Sun et al., 2016; Hu et al., 2016a). Xu et al., (2017) showed that less oxidized-OOA (LO-OOA) was mainly formed through photochemical oxidation, while the more oxidized-OOA (MO-OOA) formation was dominantly formed by aqueous-phase chemistry in Beijing for different seasons. Kuang et al. (2020) investigated the effects of gas-phase and aqueous-phase photochemical processes on the formation of SOA and found that photochemical aqueous-phase SOA formation dominantly contributed to daytime OOA formation in winter Gucheng, located between Beijing ($\sim$100 km) and Baoding ($\sim$40 km) on the NCP. We found that photochemical processing attributed mostly to MO-OOA in summertime Beijing (Gu et al., 2020). Although these studies provided important insights into SOA formation processes, our understanding on the photochemical and aqueous-phase formation pathways for SOA and their

impacts on oxidation degree are far from complete. This lack of understanding is especially so
under the conditions that atmospheric oxidative capacity and pollution characteristics have been
largely changing in China due to large reduction in direct emissions of air pollutants.
In this study, we investigated the photochemical versus aqueous-phase processing for SOA
composition and oxidation degree of OA in summertime Handan, which is a typical
industrialized city in the NCP region. The city is located at the intersectional area of Hebei,
Shanxi, Henan, and Shandong–four heavily urbanized and industrialized provinces (Fig. S1),
and it is therefore an ideal site to investigate the SOA formation pathways in the NCP region.
The multiple formation pathways, evolution of SOA composition, and oxidation degree under
different meteorological conditions were discussed, which sheds light on the aqueous-phase
chemistry and photochemical processing in SOA formation in the NCP region of China.

## 2. Experimental methods

### 2.1 Sampling site

Measurements were conducted from $10^{th}$ August 2019 to $17^{th}$ September 2019 on the campus
of Hebei University of Engineering (36.57 °N, 114.50 °E), located at the southeast edge of urban
Handan (Fig. S1). The site is surrounded by a school and residential areas, ~300 m north to
South Ring Road and ~400 m northeast to the Handan Highway (S313). The sampling site is
on the rooftop of a four-floor building, approximately 12 m above the ground.

### 2.2 Instrumentation

Real-time non-refractory $PM_{2.5}$ composition was measured by a soot particle long time-of-
flight aerosol mass spectrometer (SP-LToF-AMS, Aerodyne Research Inc.) with a time
resolution of 1 min. The detailed instrument description and operation of AMS were reported
in Onasch et al., (2012). Compared to the conventional AMS, the LToF mass analyzer can
provide much better mass resolution of ∼8000. During the campaign, the instrument was
operated in the "laser off" mode and only the standard tungsten vaporizer was applied.
Therefore, only non-refractory $PM_{2.5}$ components (NR-$PM_{2.5}$) were measured, including
organics (Org), nitrate (NO3), sulfate (SO4), ammonium (NH4), and chloride (Chl). Ambient
air was sampled and dried by a Nafion dryer (MD-700-24S, Perma Pure, Inc.) at a flow rate of
5 L $min^{-1}$, and then sub-sampled into the SP-LToF-AMS at a flow rate of ~ 0.1 L $min^{-1}$. An
aerodynamic $PM_{2.5}$ lens was used to focus the particle into a beam, which was then impacted
on the heated tungsten surface (~ 600 ℃) and flash-vaporized. Electron ionization with 70 eV
was used to ionize the vaporized gases. The ionization efficiency (IE) and the relative ionization
efficiency (RIE) calibrations (Jimenez et al., 2003) were conducted by using 350 nm
ammonium nitrate ($NH_4NO_3$) and ammonium sulfate (($NH_4$)$_2SO_4$) particles.
Gaseous pollutants including $SO_2$ (9850 $SO_2$ analyzer, Ecotech), $NO_2$ (Model 42i NO-NO$_2$-
$NO_x$ analyzer, Thermo Scientific), CO (Model 48i carbon monoxide analyzer, Thermo
Scientific), $O_3$ (Model 49i ozone analyzer, Thermo Scientific), and meteorological parameters
including RH and temperature were also measured during the observation period. Furthermore,
an aethalometer (Model AE-33, Magee Scientific) was deployed to measure the mass
concentration of black carbon (BC) at a time resolution of 1 min.
**2.3 Data Analysis**
The data analysis software (SQUIRREL, version 1.63I and PIKA, 1.23I) within Igor Pro 6.37
(WaveMetrics) was used to analyze the AMS data. The experimental RIE values of 4 (NH4)
and 1.2 (SO4) and the standard RIE values of 1.4 (Org), 1.1 (NO3) and 1.3 (Chl) were used.
The composition-dependent collection efficiency (CDCE, Middlebrook et al., 2012) was used
to compensate for the incomplete detection caused by particle bounce on the vaporizer. An
improved Ambient (I-A) method was adopted for the elemental ratio analysis of high-resolution
(HR) OA mass spectra, such as oxygen-to-carbon (O:C), and hydrogen-to-carbon (H:C) ratios
(Canagaratna et al., 2015), which reflect the relative composition and oxidation degree for
different OA source. In our study, PMF was performed on HR mass spectra of OA for ions with
*m/z* values of 12-120, together with the signals from integer *m/z* values between 121 to 300 (i.e.,
unit mass resolution, UMR) using SoFi (version 6.3) in Igor Pro (Paatero, 1999; Canonaco et
al., 2013). The data and error matrices were preprocessed according to Elser et al., (2016) and
detailed description of PMF analysis was given elsewhere (Canonaco et al. 2013; Elser et al
2016). Unconstrained PMF solutions with varied factor numbers were analyzed and six factors
were resolved, including two primary and four secondary organic factors (Fig. 3). The six-factor
solution was preferred because the five-factor solution was not able to separate high signal of
*m/z* 44 (which represents high oxidation state) from primary organic aerosol (POA) factors,
while the seven-factor solution added additional OOA factors with similar profiles and noisy
time series for which no physical interpretation could be found. The two POA factors consisted
of a traffic-related factor (hydrocarbon-like OA, HOA) and a cooking-related factor (COA),
which had been resolved in previous summer studies in NCP (Elser et al., 2016; Hu et al., 2016b;
Sun et al., 2016; Huang et al., 2019). AMS source apportionment studies often report one or
two oxygenated organic aerosol (OOA) factors that are distinguished by the extent of
oxygenation and linked to volatility or oxidation degree. Owing to higher mass resolution of
LTOF-AMS and the inclusion of integer-mass signals for *m/z* from 121 to 300 for high-
molecular-weight species such as polycyclic aromatic hydrocarbons (PAHs), we herein
resolved four SOA factors. These four SOA factors include aq-SOA attributable to aqueous-
phase chemistry, $O_x$-initiated-SOA attributable to photochemistry, primary-related-SOA
attributable to prompt oxidation of POA during emission, and fresh-SOA with a lower $f_{44}/f_{43}$
ratio (fraction of *m/z* 44 and 43 in OA).

**2.4 Aerosol liquid water content**

The aerosol liquid water content (ALWC) was simulated by ISORROPIA-II model (Fountoukis and Nenes, 2007; Hennigan et al., 2015) using the measurements of ambient inorganic species (NO3, SO4, NH4, and Chl) and meteorological parameters (temperature and RH). The simulation was run in "metastable" mode where all components are assumed to be deliquescent and contain no solid matter. The concentrations and speciation (if dissociated) of those inorganic species in thermodynamic equilibrium was then simulated by the model and then the ALWC was calculated. The inorganic cations such as $Na^+$, $K^+$, $Ca^{2+}$, $Mg^{2+}$ were not measured and included in the simulation on account of that these crustal ions constituted relatively small fractions of aerosol, and had relatively weak effects on ALWC accumulation (Fountoukis and Nenes, 2007;Su et al., 2022). The ISORROPIA-II model does not consider the contribution to ALWC from organics, since inorganic aerosols dominate the water uptake by ambient particles with a contribution of approximate >80% of the total ALWC (Huang et al., 2020).

**3. Results and discussion**

**3.1 SOA sources**

In our study, SOA accounted for 69% (13.5 $\mu g$ $m^{-3}$) of the total OA (19.6 $\mu g$ $m^{-3}$), representing the dominant fraction in OA in summer Handan. Among the four PMF-resolved SOA sources (Fig. 1), $O_x$-initiated-SOA dominated (31% to total OA), followed by fresh-SOA (18%), aq-SOA (15%), and primary-related-SOA (5%). Since we focus on SOA formation in this study, detailed descriptions of the HOA (12%) and COA (19%) is provided in section 1.1 in the SI. The mass spectral profiles of the six OA source factors are shown in Fig. 1, while the time series of the SOA factors are shown in Fig. 2. In particular, a remarkable continuous growth of aq-SOA concentration (from ~0.3 $\mu g$ $m^{-3}$ to 25.2 $\mu g$ $m^{-3}$) and ALWC (from 3.1 $\mu g$ $m^{-3}$ to 486.1 $\mu g$ $m^{-3}$) occurred on $24^{th}$-$28^{th}$ August (Fig. 2d). Meanwhile, the O:C ratio indicative of OA oxidation state displayed a continuous increase from 0.52 to a maximum of 0.93 during this time (Fig. 2e), consistent with the continuous increase in RH (reaching over 95%). This observation hints that during this period aqueous-phase processing might have played an important role in aq-SOA formation. This role of aqueous-phase processing in SOA formation is not just specific to this particular event, but also important in the whole campaign, which is discussed in detail in section 3.3 later.

SOA factors were resolved depending on the oxidation state, which correspond to aged SOA and fresh SOA respectively (Jimenez et al., 2009). One factor is attributed to aqueous-phase chemistry (aq-SOA) and the other to photochemistry ($O_x$-initiated-SOA), while fresher factor is produced by fresh-source (fresh-SOA) with a lower $f_{44}/f_{43}$ ratio, and the other considered as

oxidized primary sources denoted as primary-related-SOA. Although all of the SOA factors
were characterized by higher $m/z$ 44 ($CO_2^+$) and $m/z$ 28 ($CO^+$) signal compared with POA
factors, their mass spectrum and temporal trends were noticeably distinguishable,
corresponding to different formation mechanism, which will be discussed in the following
section.
As shown in Fig. S3, the aq-SOA was identified as it increased with ALWC but decreased
with $O_x$, which might be produced/influenced by aqueous-phase chemistry and is defined as aq-
SOA. This indicates that aq-SOA was either formed via aqueous phase reactions or
absorbed/dissolved into aerosol liquid water. It exhibits the highest O:C ratios of all factors (0.7)
and a higher $f_{CO2+}$ to the total signal of 21.7%, but a low H:C ratio of 1.24 (Fig. 1). The $O_x$-
initiated-SOA in our study is photochemical production SOA whose formation initiated with
the presence of $O_x$. As $O_x$ has been shown to be a conserved tracer to during photochemical
processing (Xu et al., 2017), the relationship between $O_x$ and $O_x$-initiated-SOA can represent a
metric to characterize SOA formation mechanisms associated with ozone production chemistry
SOA (Herndon et al., 2008). $O_x$-initiated-SOA presented an opposite trend with significant
increase as function of $O_x$ but decreased as a function of ALWC (Fig. S3), suggesting the
dominant role of photochemical processing in the formation of $O_x$-initiated-SOA.
The fresh-SOA showed an increase substantially as ALWC increasing, similar to aq-SOA.
Whereas it also showed a slight increase trend following $O_x$ when $O_x <$ 100 ppb (Fig. S3).
Therefore, both aqueous-phase chemistry and photochemical processing were thought to have
positive impacts synchronously on the formation of fresh-SOA. In this study, the fresh-SOA
had the lowest atomic O:C ratio of 0.41 and the highest atomic H:C ratio of 1.41 among the
four SOA factors, corresponding with the $f_{CO2+}$ of 8.3%, these characteristics are consistent with
the global average O:C ratio of LO-OOA of 0.35 $\pm$0.14, Ng et al., 2010), demonstrating the it
is more fresh SOA. Although the primary-related-SOA constituted a small fraction and showed
little variation, this SOA source is also of particular interest because of its distinctive fragments
with high $m/z$ values in the mass spectrum (Fig. 1d). At $m/z <$ 120, the primary-related-SOA
had higher intensities for $m/z$ 43 (mainly $C_2H_3O^+$) and $m/z$ 44 (mainly $CO_2^+$) than those in POA,
indicating a typical nature of less-oxidized SOA. At $m/z >$ 120, PAH-derived fragments are
clearly evident in the mass spectrum of the primary-related-SOA, as indicated by PAH-like ions
(described in SI 1.2) (Dzepina et al., 2007). Previous AMS studies have observed pronounced
peaks of PAH ions in POA mass spectra, such as those in coal combustion organic aerosol
(CCOA) and biomass burning organic aerosol (BBOA) (Hu et al., 2016b; Zhao et al., 2019),
but rarely in SOA. This observation implies that the factor may be related to the POA originated
from domestic coal combustion and here it is termed as primary-related-SOA (Xu et al., 2006).
Moreover, this SOA factor exhibited relatively better correlations with some gaseous pollutants
(Fig. S4), such as CO ($R = 0.6$) and $NO_2$ ($R = 0.5$), and also tracked with HOA ($R = 0.4$). These
observations suggest that the primary-related-SOA might be transformed from locally emitted
POA as a non-negligible source to SOA.
To further investigate the SOA formation mechanism, the dataset was segregated into three
periods according to different features depends on meteorological parameters (Fig. 2), i.e., the
reference period (P1), high-$O_x$ period (P2) and high-RH period (P3). Briefly, the reference
period, P1, was characterized by a low average OA concentration ($15.4 \pm 3.2$ $\mu$g m$^{-3}$) and was
mainly affected by clean air from southwest of the sampling site and precipitation activities
(Table S1). The high-$O_x$ period (P2) was featured by a high $O_x$ concentration ($65.1 \pm 20.4$ ppb),
warmer temperatures ($26.4 \pm 4.0$ °C) but lower RH ($57.7 \pm 17.5$ %). The mass loadings of OA
($19.8 \pm 4.7$ $\mu$g m$^{-3}$) and other pollutants in P2 were higher than those in P1 (Table S1). P3 was
assigned as a high-RH period because of the noticeably high RH ($83.7 \pm 12.5$ %) and high
ALWC ($95.4 \pm 114.2$ $\mu$g m$^{-3}$). Winds were weak ($<1.0$ m s$^{-1}$) throughout this period, indicative
of stagnant conditions, which facilitated pollutant accumulation and resulted in the highest
average OA concentrations ($25.0 \pm 6.2$ $\mu$g m$^{-3}$).
During the reference period (P1), SOA had the lowest contribution to OA (57%), and the $O_x$-
initiated-SOA and aq-SOA constituted 22% and 21% to total OA, respectively. For the high-
$O_x$ period (P2), enhanced SOA formation was found, with the SOA fraction increased to 71%
of the total OA. The $O_x$-initiated-SOA showed the highest mass loading of 7.3 $\mu$g m$^{-3}$ and
highest contribution of 37% to total OA. These increases suggest that high-$O_x$ condition
facilitates the production of SOA by photochemistry, making the $O_x$-initiated-SOA the major
source of SOA during P2. During the high-RH period (P3), SOA fraction continually increased,
approaching 79% in total OA, and the SOA was mainly contributed by aq-SOA and fresh-SOA.
The mass contribution of aq-SOA increased dramatically from 9% to total OA during P2 to 33%
during P3 (Fig. S2), and average mass concentrations from 1.8 $\mu$g m$^{-3}$ to 8.3 $\mu$g m$^{-3}$, which
suggests rapid SOA production through the aqueous-phase chemistry. Comparatively, the
contribution of fresh-SOA was about ~20% in both P2 and P3, but lower in P1 (9%), suggesting
that the formation fresh-SOA was affected by both high $O_x$ and high RH. It should also be noted
that O:C ratio increased in the succession from P1 (0.73) to P2 (0.74) and further to P3 (0.77),
accompanied by continually decrease of H:C ratio from 1.64 to 1.56, and to 1.53 (Fig. 3),
suggesting the increase of OA oxidation degree. As a result, the high $O_x$ in P2 and high RH in
P3 (as compared to P1) promoted the formation of SOA, specifically $O_x$-initiated-SOA (in P2)
and aq-SOA (in P3), leading to the increase in the degree of oxygenation in total OA.
Overall, our results suggest that SOA could be formed through different pathways, in
particular photochemistry, aqueous-phase chemistry, and conversion of POA to SOA
contributed to SOA formation.
**3.2 Photochemistry**

As expected for summertime, photochemistry associated with $O_x$ has significant impacts on the formation and evolution of SOA. Herein, the relationships between OA factors and $O_x$ were investigated to offer insights into the formation mechanisms of SOA associated with the ozone production chemistry (Herndon et al., 2008). During P2, as $O_x$ increased, the mass loadings of $O_x$-initiated-SOA showed a substantially increasing trend when $O_x$ was > 30 ppb and eventually saturated when $O_x$ was >100 ppb, raising the contribution of $O_x$-initiated-SOA from 20% to 61% of total OA (Fig. 4). This observation indicates the importance of photochemistry in the formation of $O_x$-initiated-SOA in summer, in which high $O_x$ concentration as well as temperature corresponding to strong atmospheric oxidative capacity, can accelerate the photochemical formation (Duan et al., 2021). As a comparison, the mass concentrations of other OA factors except $O_x$-initiated-SOA showed decreasing trends as $O_x$ increased (Fig. 4c). Such differences between SOA factors are likely due to the enhanced secondary production/transformation from POA and fresher SOA factors to the more aged $O_x$-initiated-SOA. Note that the O:C ratio presented a faster increasing rate as a function of $O_x$ (from 0.6 to 1.0, Fig. 4d) than those in P1 and P3, suggesting that photochemistry might result in higher OA oxidation state during P2.

The typical episode with high-$O_x$ period (P2) was dominated by a series of daytime photochemical evolutions. To evaluate the relative contributions of photochemical and aqueous-phase processing production and the transformation of these SOA factors in different meteorological stages, the average diurnal variations of OA factors, O:C ratios, $O_x$, temperature, AWLC and primary gas pollutants during different periods are shown for comparison. Fig. 6 shows that $O_x$ increased rapidly from 6:00 to 14:00 in all periods, but was highest in P2. Correspondingly, a lower mean value of ALWC (8.4 $\mu$g m$^{-3}$) was also observed in P2 than in P1 and P3. During P2, $O_x$-initiated-SOA was produced quickly and played the dominant role during daytime, while its concentration typically decreased during nighttime. The average concentration of $O_x$-initiated-SOA increased continually from 4.2 $\mu$g m$^{-3}$ at 7:00 local time (LT) to 10.4 $\mu$g m$^{-3}$ at 15:00 LT in 8 h, with the maximum $O_x$-initiated-SOA mass fraction in OA reaching 65% at 15:00 LT (Fig. S6c). This high average growth rate of 0.8 $\mu$g m$^{-3}$ h$^{-1}$ in $O_x$-initiated-SOA corresponded to the high $O_x$ concentration, high temperature and strong solar radiation in daytime, suggesting enhanced photochemistry reaction. In contrast, the concentrations and the contributions of other SOA factors deceased continuously at the same time (Fig. 6). The opposite trends between $O_x$-initiated-SOA and other OA factors from 7:00 to 15:00 LT suggest that some part of POA and fresh-SOA may convert to $O_x$-initiated-SOA by photochemical oxidation. This conclusion is consistent with findings reported by Li et al., (2020) in urban Beijing, where less-oxidized SOA may transform to more-oxidized SOA through photochemical processing as well. The O:C ratio of OA presented a significant increasingly diurnal variation with a noon peak around 14:00 ~ 16:00 LT in P2, which had the

highest value of 0.74 compared with P1 and P3, suggesting the potential transformation from
POA factors and fresh SOA factors to $O_x$-initiated-SOA could also noticeably affect OA
characteristics such as oxidation state in summer daytime. It is further indicated by a small
afternoon peak of the more oxidized tracer $CO_2^+$ (*m/z* 44) and the decrease in a less oxidized
tracer $C_2H_3O^+$ (*m/z* 43) (Fig. 7b). As a result, the mass spectra, which were initially fresh SOA
products evolved to become aged SOA products as the photochemical age increased. Overall,
with little water in the particles, the high solar radiation and high $O_x$ concentration during
daytime associated with a relatively high degree of oxygenation of OA suggest that gas-phase
oxidation and partitioning processes are probably the dominating process in SOA formation
during P2.
In addition, these results further support the idea that during the high-$O_x$ period of summer,
photochemistry has significant impacts on SOA formation, especially on $O_x$-initiated-SOA.
Note that the role of photochemistry in the formation of $O_x$-initiated-SOA is not only limited to
the gas-phase photochemistry, but also can also occur in the aqueous phase (Kuang et al., 2020).
This is the case for P3 in our study, which is discussed further in section 3.3 below.
**3.3 Aqueous-phase chemistry**
The aqueous-phase chemistry has imposed significant impacts on SOA formation during this
field campaign. To further explore the formation mechanism of SOA associated with aqueous-
phase chemistry, the relationships between different OA factors and ALWC were investigated.
During P3, the mass concentration of aq-SOA increased from 5 $\mu$g m$^{-3}$ to 17 $\mu$g m$^{-3}$, yet its
fraction showed a particularly pronounced rise from 22.5% to 52% of total OA when ALWC
increased from 0.3 to 200 $\mu$g m$^{-3}$ (Fig. 5e and f). Note that there are still consistent mass
concentrations of aq-SOA even when ALWC is very low (data interval ranging from 0~40 $\mu$g
m$^{-3}$), which is due to that over 80% of ALWC mass concentration were loaded in the first
interval, leading to a higher mean value of aq-SOA mass concentration. Actually ALWC
showed quite low mass loading in most period time but increased dramatically during P3, yet
the time series of aq-SOA and ALWC were remarkably well correlated throughout the entire
campaign ($R$=0.7, Fig. S4) rather than a strong correlation observed only in P3. This general
correlation further confirms the important role of aqueous-phase chemistry in the formation of
aq-SOA and characterized the aqueous-phase formation of aq-SOA throughout the campaign
rather than only in the high-RH event as shown in section 3.1 earlier. We also found that the
concentration and fraction of aq-SOA became stable when ALWC was > 200 $\mu$g m$^{-3}$, which is
probably attributable to that the aq-SOA formation within droplets was soon outweighed by the
scavenging processes when RH was high enough (> 95%). The O:C ratio shows an obvious
increase from 0.7 to around 0.85 when ALWC increases to 200 $\mu$g m$^{-3}$, after which it remains
relatively stable (0.85) as the ALWC increases further (Fig. 5). These results suggest that
aqueous-phase chemistry can affect the oxidation degree of OA by changing SOA composition,
especially the enhanced contribution of aq-SOA. However, the growth rate of O:C ratios as
ALWC increases in P3 was lower than that in P2 (up to 1 as $O_x$ increases). Also, the correlation
between O:C vs. $O_x$ in P2 ($R$=0.6) was stronger than O:C vs. ALWC ($R$=0.3) (Fig. S8).
Fig. 6 illustrate the different types of aqueous-phase chemistry in daytime and nighttime.
During the nighttime in P3, aqueous-phase oxidation was also enhanced during nighttime
(19:00–07:00 LT). As shown in Fig. 6, O:C ratio (0.76) at nighttime in P3 was higher than those
in P2, while exhibiting a much smaller peak during daytime. Compared with the low ALWC in
P2, the much higher ALWC concentration (peak value of 235.9 $\mu$g m$^{-3}$ at 2:00 LT) and higher
RH (peak value of 93.7% at 6:00 LT) during nighttime in P3 suggested a dominant contribution
by aqueous-phase processing. The aq-SOA shows a quite clear and unique diurnal pattern in
P3, with much higher mass concentration during the whole day (especially at nighttime) than
those in P1 and P2. After 17:00 LT, aq-SOA started to increase from 4.7 $\mu$g m$^{-3}$ to 12.7 $\mu$g m$^{-3}$
at 7:00 LT, which showed a rapid nighttime growth rate of 0.6 $\mu$g m$^{-3}$ h$^{-1}$, indicating enhanced
SOA formation through aqueous-phase chemistry at night. Whereas $O_x$-initiated-SOA
decreased rapidly from 8.2 $\mu$g m$^{-3}$ at 17:00 LT until reaching its lowest concentration of 2.6
$\mu$g m$^{-3}$ until the morning, suggesting the gas-to-particle partitioning at night under high ALWC
conditions. Furthermore, this transformation could be supported by the increase in $CO_2^+$ ($m/z$
44) and the decrease in a less oxidized tracer $C_2H_3O^+$ ($m/z$ 43) at night (Fig. 7c). Since when
the ALWC is sufficiently high, it was likely to accommodate much of the precursor organics
and oxidants to low-volatility products through aqueous-phase oxidation. In addition, the dark
aqueous-phase SOA formation was likely strong enough to counteract the nighttime scavenging
processes under high-RH conditions. Therefore, the dark aqueous-phase chemistry forming aq-
SOA shows a dominant role (over 40% to OA) during nighttime in P3.
However, during the daytime, the mass concentration of aq-SOA decreased from 7:00 to
17:00 LT in P3, coinciding an obvious increase trend of $O_x$-initiated-SOA at the same time with
an average growth rate of 0.6 $\mu$g m$^{-3}$ h$^{-1}$ (Fig. 6). This phenomenon suggests photochemical
processing can also occur in the aqueous phase when RH and ALWC were still high.
Photochemical reactions through both aqueous-phase and gas-phase can contribute
substantially to the SOA formation in polluted areas of NCP, and during haze days with high
RH and ALWC the aqueous-phase photochemical processes played a dominant role in daytime
SOA formation (Kuang et al., 2020). The rapid daytime $O_x$-initiated-SOA formation in our
study possibly occurred on the particle surface and in the aerosol liquid water (Ervens et al.,
2011) under humid conditions with high ALWC but driven by gas-phase direct photolysis and
oxidation by photooxidants under high $O_x$ conditions. Under such high-RH level (RH > 80%),
the water-soluble species produced from photochemistry in the gas phase can also partition into
the aqueous phase and be further oxidized to form low-volatility products (Carlton et al., 2007;
Sullivan et al., 2016). Previous studies have demonstrated that gas-phase oxidants such as OH
radicals and $H_2O_2$ can also partition to the aqueous phase to further oxidize dissolved the
oxidized VOCs (OVOCs) into aq-SOA (Ye et al., 2018). Other studies also revealed that
photochemical reactions in the aqueous droplets can occur through direct photolysis or through
oxidation by oxidants (Ervens et al., 2011; 2014;Ye et al., 2018). Therefore, in our campaign,
dark aqueous-phase chemistry is responsible for rapid aq-SOA formation during nighttime,
while the aqueous-phase photochemistry during daytime is likely prevail by rapid daytime $O_x$-
initiated-SOA formation during P3. This comparison demonstrates that the nocturnal aqueous-
phase chemistry and daytime aqueous-phase photochemistry are both important pathways in
the total SOA growth. The aqueous-phase chemistry related to fresh-SOA is more complicated,
requiring both daytime radiative conditions and certain amounts of ALWC in nighttime. For
example, Fig. 5e shows that the fresh-SOA has a similar increasing trend with aq-SOA as
ALWC increased, however, it also increased slightly as $O_x$ increased (Fig. 4e), hinting that both
ALWC and the oxidants are critical for fresh-SOA formation and both the aqueous-phase
chemistry and the photochemistry (including that in the aqueous phase) participated to produce
fresh-SOA simultaneously. It is worth noting that three peaks were found in the diurnal
variation of fresh-SOA in P3. The peaks at around 6:00 and 19:00 LT at night were similar to
those of aq-SOA and lower than it, while the peak at around 13:00 LT is consistent with the
peak in the diurnal cycle of $O_x$ (Fig. 6). Although there is also a smaller peak around 13:00 LT
in P3, the whole pattern of aq-SOA is characterized by decreasing trend at daytime. These
results suggest that fresh-SOA could be formed through dark nighttime aqueous-phase reactions,
which are partially reversible upon the evaporation of aerosol liquid water, and also formed
through photochemical aqueous-phase reactions during daytime. Different from aq-SOA,
which is highly correlated and limited with ALWC, two types of aqueous-phase chemistry in
daytime and nighttime are dominant pathways to the fresh-SOA growth. Our analysis on
formation pathways of these SOA factors suggested the potential interactive roles of gas-phase
oxidation, gas-particle partitioning, and aqueous-phase oxidation in the formation of SOA.
**3.4 SOA from POA transformation**
The photochemistry and aqueous-phase chemistry show distinct effects on POA evolution
and SOA formation. The relationships between $O_x$-initiated-SOA /aq-SOA and other POA-
related components (HOA + COA + primary-related-SOA) were plotted in Fig. S9. A strong
negative correlation ($R$=-0.8) between POA-related components and   $O_x$-initiated-SOA was
observed (Fig. S9c), consistent with the decrease in mass concentration of POA-related
components during P2. This observation suggests that the production of $O_x$-initiated-SOA was
at least partly facilitated by photochemical transformation of other OA components. However,
the better diffusion conditions in P2 might also attribute a great extent to the negative
correlation, as the formation period of $O_x$-initiated-SOA usually occurred during the noontime
when the boundary layer was much developed, while the POA usually decreased via horizontal
and vertical diffusion. In comparison, POA-related components and aq-SOA correlate weakly.
When ALWC ($<20$ $\mu$g m$^{-3}$) and nitrate concentrations were lower ($< 3$ $\mu$g m$^{-3}$), mostly during
P1 and P2, POA-related components and aq-SOA showed almost no correlation ($R$=0.1and $R$=-
0.1). However, when ALWC concentration and nitrate concentration were higher than those
thresholds above (data points with yellow/red colors mostly during P3), they had a relatively
good negative correlation ($R$=-0.5) (Fig. S9f), indicating the importance of ALWC and nitrate
in aqueous-phase chemistry. This is consistent with results in winter Beijing (Wang et al., 2021),
where POA factor had strong negative correlations with aq-SOA, suggesting that these POA
factors might produce aq-SOA by aqueous-phase oxidation. In addition, under high-ALWC
conditions, nitrate had similar formation mechanisms with aq-SOA or high nitrate supports the
potential formation/transformation from POA-related components to aq-SOA, which is
consistent with the results in section 3.3. The phenomenon of negative correlation between
POA-related components and SOA at high $O_x$/ALWC further emphasizes the importance of
conversion from POA to SOA.
As shown in the Van Krevelen (VK) plot (Fig. 8a), O:C and H:C both increase in the
succession from primary-related-SOA to $O_x$-initiated-SOA and eventually to aq-SOA,
supporting a successive oxidation sequence from primary-related-SOA to aq-SOA. Generally,
H:C shows a decreasing trend as O:C increases for organic compounds during oxidation in
other studies (Ng et al., 2011; Gilardoni et al., 2016; Lee et al., 2017: Zhao et al., 2019; Chen
et al., 2021), suggesting a general negative correlation between H:C and O:C. This positive
relationship of O:C and H:C evolution during oxidative aging in this study is interesting. It
might be caused by ring-breaking reactions which could further promote the transformation of
aromatic POA to aq-SOA. Previous studies in both laboratory (Huang et al., 2018; Wang et al.,
2020) and field (Hu et al., 2016a) demonstrated that the OH-initiated ring-breaking reactions
of aromatic species can occur in the aqueous phase and form highly oxidized oxygenated
compounds. For example, Hems and Abbatt (2018) suggested that nitrophenol molecules could
react rapidly with OH radicals in aqueous solutions with the addition of OH functional groups
to the aromatic ring at the initial stage, followed by fragmentation to multifunctional organic
species with high H:C and O:C ratios. Wang et al. (2021) found that the ring-breaking oxidation
of aromatic FF-POA was the mechanism for aq-SOA formation. Similar to those in primary-
related-SOA, PAH-like ions was also found in the mass spectrum of aq-SOA at $m/z > 150$,
albeit less pronounced, consistent with a previous study in Beijing (Wang et al., 2021). This is
likely due to the oxidation of PAHs in the conversion of primary-related-SOA and aq-SOA,
which is caused by enhanced hydroxylation of the aromatic ring and increased yields of
carboxylic acids in OH-initiated reactions (Sun et al., 2010). This kind of ring-breaking
oxidation of aromatic POA could thus lead to aq-SOA formation (Huang et al., 2018; Wang et
al., 2021). In addition, the locations of aq-SOA and the slope of overall OA are near the line
with the slope of -1 in the VK plot, indicating more carboxylic acid formation while the
replacement of a hydrogen atom with a carboxylic acid group ($-COOH$) (Heald et al., 2010;
Ng et al., 2011). This observation supports that oxidation of PAHs was probably involved in
the conversion of primary-related-SOA to aq-SOA through aqueous-phase chemistry, leading
to functionalization as carbonyls and carboxylic acids.
Specifically, the organic fragments and mass spectrum evolution of OA were analyzed to
illuminate the transformation in photochemical processing and aqueous-phase chemistry. Fig.
8b shows the mass fractions of $CH_2O_2^+$, $CH_3SO^+$, $HCO_2^+$, and $C_2H_2O_2^+$ ion fragments in OA as
a function of ALWC. The aq-SOA was tightly correlated with $CH_2O_2^+$ ($R^2 = 0.81$) at $m/z$ 46
and $CH_3SO$ ($R^2 = 0.78$) at $m/z$ 63 (Fig. S10), Consistently, both of them showed increase trends
as ALWC increasing, similar as aq-SOA, which indicating typical fragment characteristics of
ions of aqueous-phase processing products (Tan et al., 2009; Sun et al., 2016; Duan et al., 2021).
The intensities of $HCO_2^+$ ($m/z$ 45), a common fragment ion of carboxylic acids, is associated
with aqueous oxidation of aromatic compounds. $C_2H_2O_2^+$ ($m/z$ 58) is a tracer ion for glyoxal,
which could be a ring-breaking product from the aqueous-phase oxidation of PAHs. The
increasing trends of these ions with ALWC suggest that water-soluble organic species such as
carboxylic acids and glyoxal are produced as components of aq-SOA following aromatic
oxidation and ring breaking. Moreover, the concentration of PAHs increased with the increase
of ALWC (Fig. S11), consistent with the oxidation of PAHs from ring-breaking reactions that
can take place in the aqueous phase and being involved in the conversion to aq-SOA.
**4. Conclusion**
The sources and formation mechanisms of SOA were investigated by online aerosol mass
spectrometry and statistical (PMF) analysis from August to September of 2019 in Handan, a
mid-sized industrialized city in NCP of China. Four specific SOA factors were resolved,
including aq-SOA (15% to total OA), $O_x$-initiated-SOA (31%), fresh-SOA (18%) and primary-
related-SOA (5%). By studying the formation of these SOA factors in different selected periods
(P1-P3) against $O_x$ and ALWC, we found multiple pathways leading to their formation,
sometimes with mixed pathways for one type of SOA.
Both photochemistry and aqueous-phase chemistry resulted in enhanced OA oxidation state.
During high-$O_x$ period, photochemistry had imposed significant impacts on the formation and
evolution of SOA in summertime. The $O_x$-initiated-SOA contributed up to 65% to total OA in
the daytime, with a high average growth rate of $0.8\,\mu\mathrm{g\,m^{-3}\,h^{-1}}$, suggesting the efficient daytime
formation of SOA from photochemistry. Rapid increases of the concentration and contribution
(up to 61%) of $O_x$-initiated-SOA were found as $O_x$ increased, while all the other OA factors
showed decreasing trends with $O_x$ concentration increasing. The difference suggests enhanced
secondary transformation from POA/fresh SOA factors to the more aged $O_x$-initiated-SOA
under high-$O_x$ condition. However, during the high-RH period, two types of aqueous-phase
chemistry were both important pathways for the SOA growth. During nighttime and under high-
RH conditions, dark aqueous-phase chemistry played significant roles with rapid aq-SOA
formation (up to 45% in total OA), while the aqueous-phase photochemistry was more
important by rapid $O_x$-initiated-SOA formation during daytime (up to 39% in total OA). The
primary-related-SOA was evidently linked to the POA originated from coal combustion
activities, as indicated by the PAH-like ion peaks. Although it constituted a small fraction of
5%, the potential transformation and conversion from primary-related-SOA to aq-SOA could
also be an important pathway via hydroxylation of the aromatic ring or ring-breaking oxidation
of aromatic POA species through aqueous-phase chemistry. This study highlights the multiple
reaction pathways, on top of multiple precursor types, on the SOA formation in industrialized
regions, and calls form more in-depth study on the interactive roles of those formation pathways.

*Data availability*. Raw data used in this study are archived at the Institute of Earth Environment,
Chinese Academy of Sciences, and are available on request by contacting the corresponding
author.
*Supplement.* The Supplement related to this article is available online.
*Competing interests.* The authors declare that they have no conflict of interest.
*Author contributions.* RJH designed the study. Data analysis and source apportionment were
done by YFG and RJH. YFG and RJH wrote the manuscript. YFG and RJH interpreted data
and prepared display items. All authors commented on and discussed the manuscript.
**Acknowledgement**
This work was supported by the National Natural Science Foundation of China (no.
41925015), the Key Research Program of Frontier Sciences from the Chinese Academy of
Sciences (no. ZDBS-LY-DQC001), the Strategic Priority Research Program of the Chinese
Academy of Sciences (no. XDB40000000), and SKLLQG (no. SKLLQGTD1801).

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

**Figures**

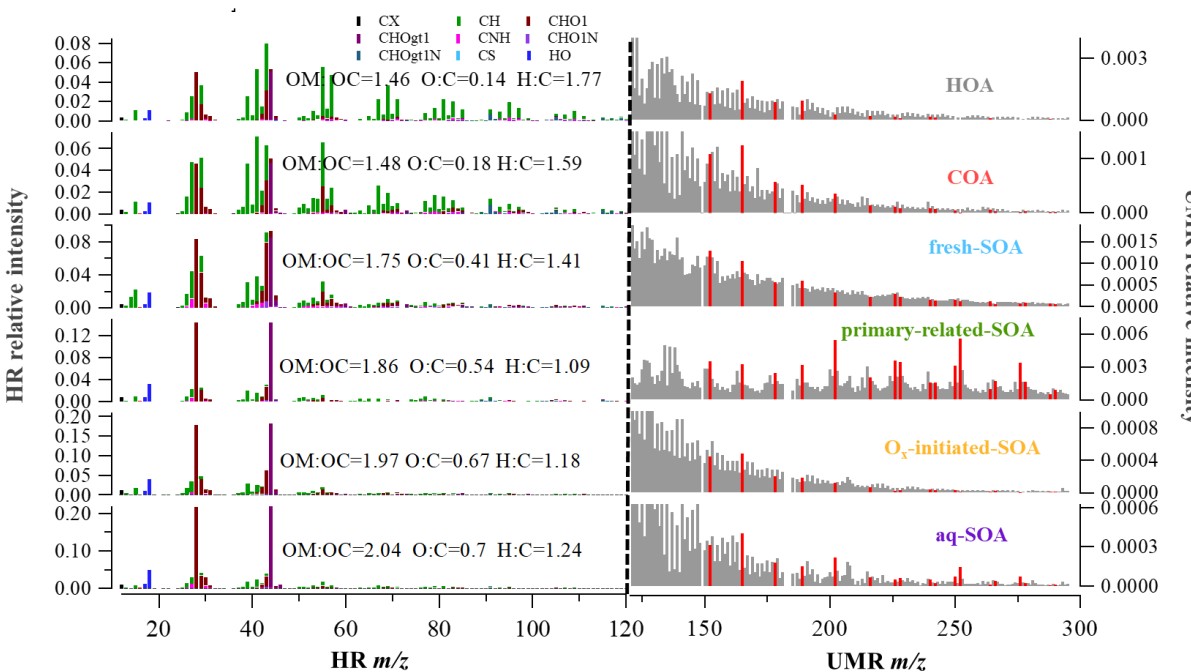


**Fig. 1** HR and UMR mass spectra of OA factors: (a) HOA; (b) COA; (c) fresh-SOA; (d)
primary-related-SOA; (e) $O_x$-initiated-SOA; (f) aq-SOA. Mass spectra signals less than 120
amu are colored by nine ion categories, signals equal to or greater than 120 amu are in unit
mass resolution, and polycyclic aromatic hydrocarbons (PAHs) signals are in red on the right
panels.


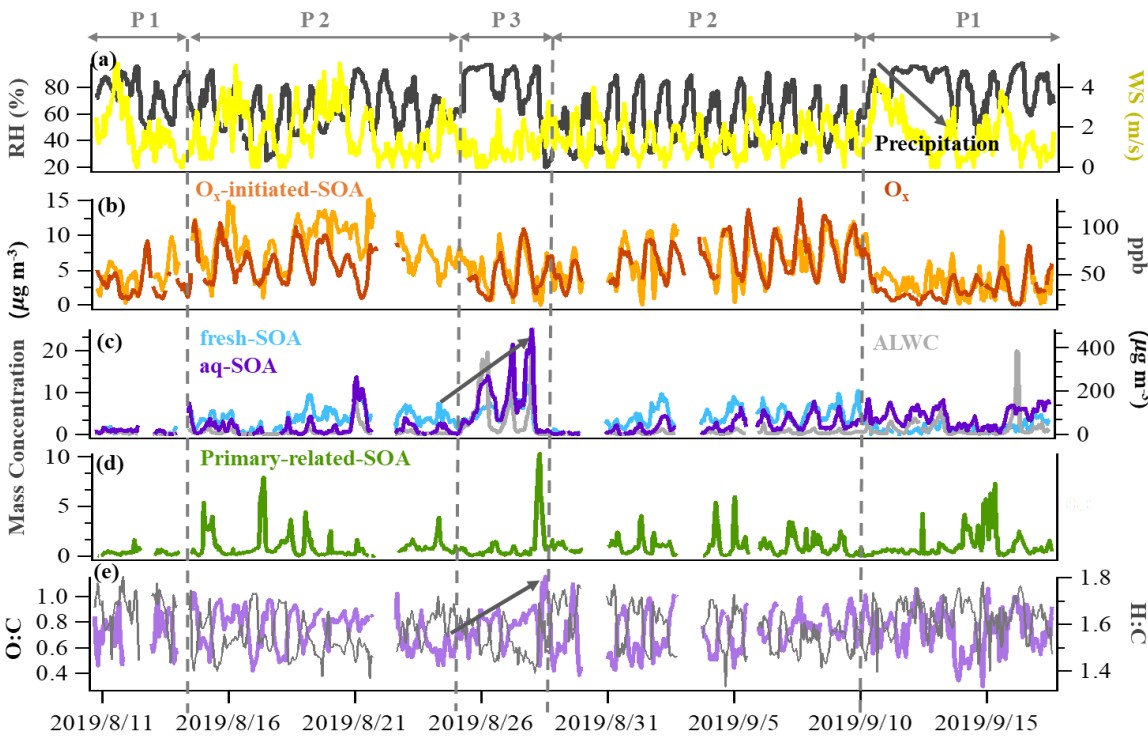


**Fig. 2** Time series of (a) relative humidity (RH) and wind speed (WS), (b) $O_x$ and $O_x$-initiated-SOA, (c) fresh-SOA, aq-SOA and ALWC, (d) primary-related-SOA, (e) the O:C ratio and H:C ratio. The time series were categorized to be three typical periods based on total SOA mass concentrations and meteorology conditions: reference period (P1), high $O_x$ period (P2) and high RH period (P3).


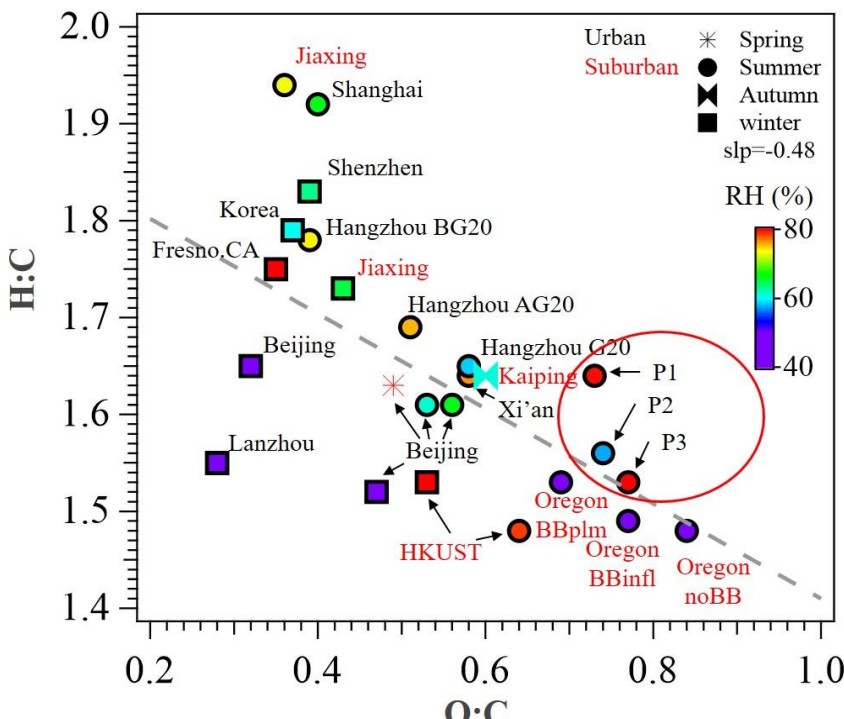


**Fig.3** Van Krevelen plot for OA of urban and suburban sites in China and other nations. Data points are colored by RH (%). P1, P2 and P3 in red circles represents the different periods in this study. All the data and related references can be found in Table S3.

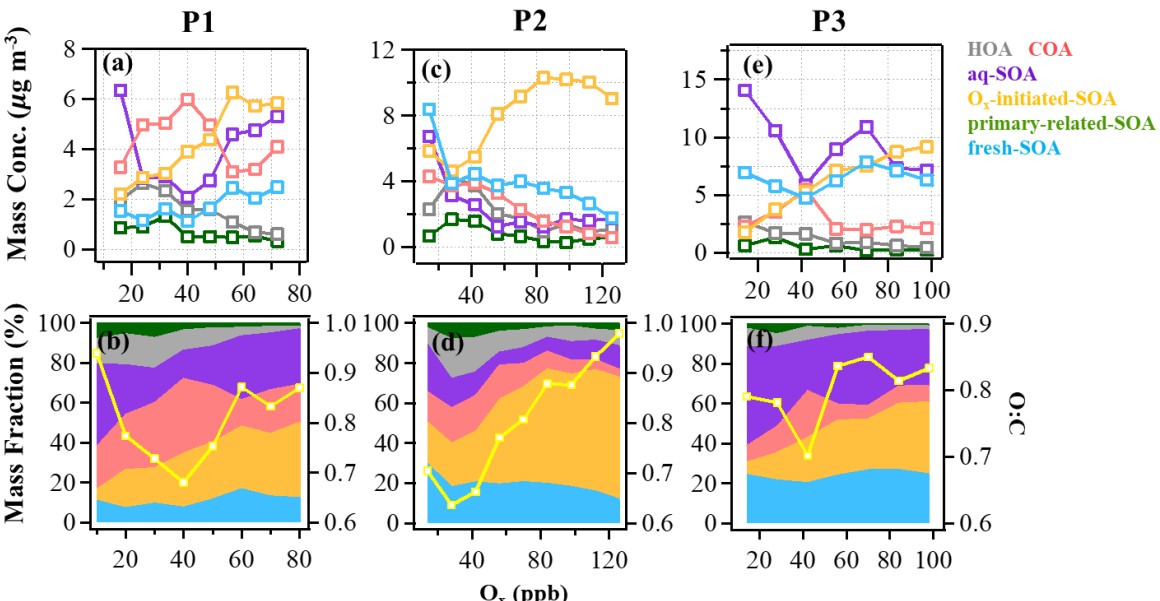


**Fig. 4** The mass concentration and contribution of OA factors as functions of $O_x$ in reference period (P1: a & b), high $O_x$ period (P2: c & d) and high RH period (P3: e & f) during this

campaign. The yellow curves represent the O:C ration vs. $O_x$. The data were binned according
to $O_x$ concentration (10 ppb increment in P1, 14 ppb increment in P2 and P3.

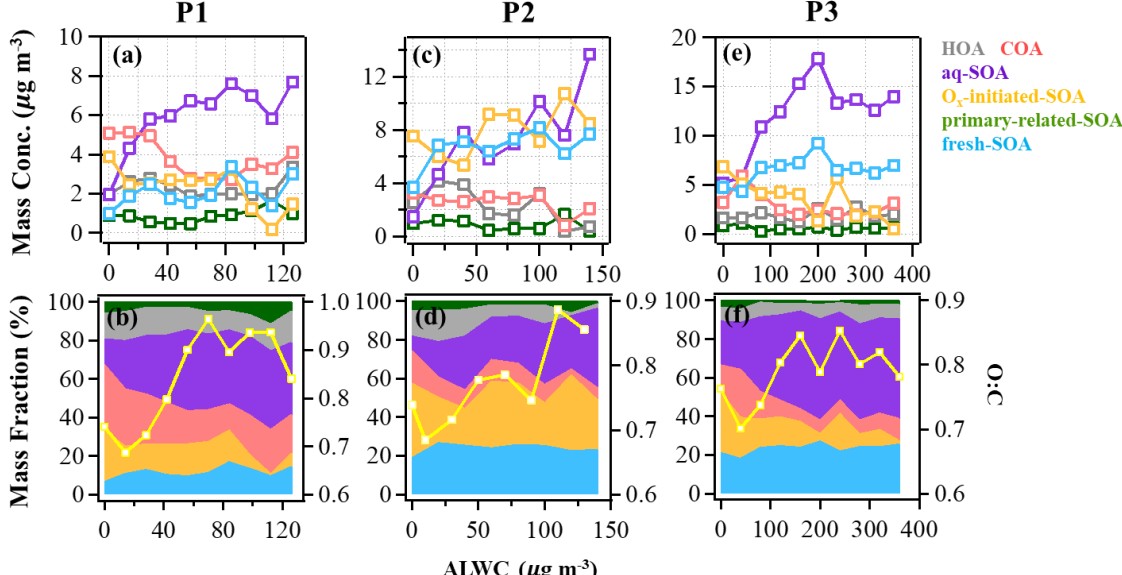


**Fig. 5** The mass concentration and contribution of OA factors as functions of ALWC in reference period (P1: a & b), high $O_x$ period (P2: c & d) and high RH period (P3: e & f) during this campaign. The yellow curves represent the O:C ration v.s. ALWC. The data were binned according to the ALWC concentration (14 $\mu gm^{-3}$, 20 $\mu gm^{-3}$ and 40 $\mu gm^{-3}$ increment in P1 P2 and in P3).

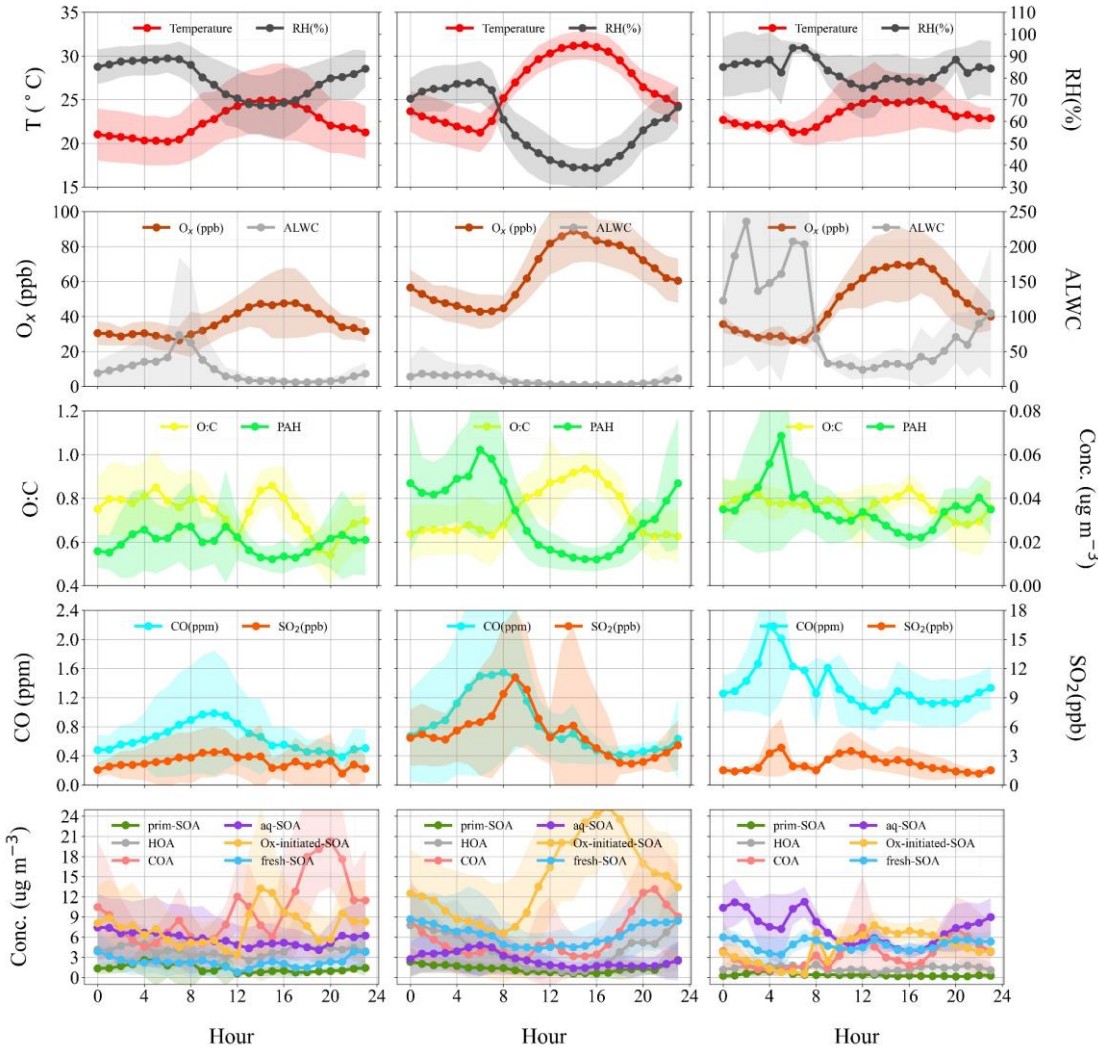


**Fig. 6** Diurnal patterns of meteorological parameters (T, RH), gaseous species ($O_x$, CO, $SO_2$),
ALWC (liquid water content), O:C (oxygen-to-carbon elemental ratio), polycyclic aromatic
hydrocarbons (PAHs) fragments and OA factors in reference period (P1), high $O_x$ period (P2)
and high RH period (P3) in this campaign.

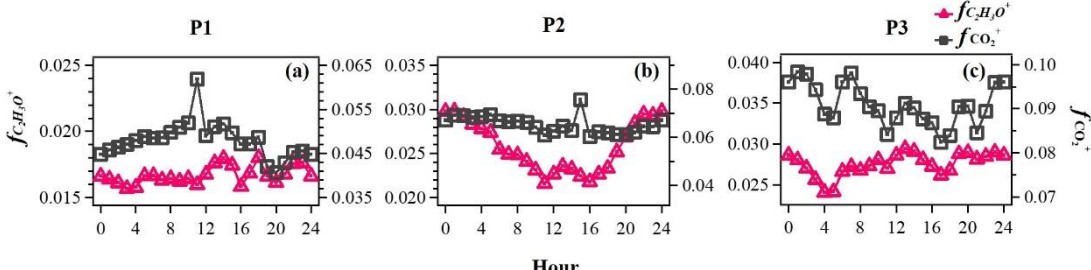


Fig. 7 Evolution of high-resolution organic mass spectra on changes in relative intensities (mass fraction) of oxygen-containing ions: $C_2H_3O^+$ ($m/z$ 43) and $CO_2^+$ ($m/z$ 44) in reference period (P1: a), high $O_x$ period (P2: b) and high RH period (P3: c) in this campaign.

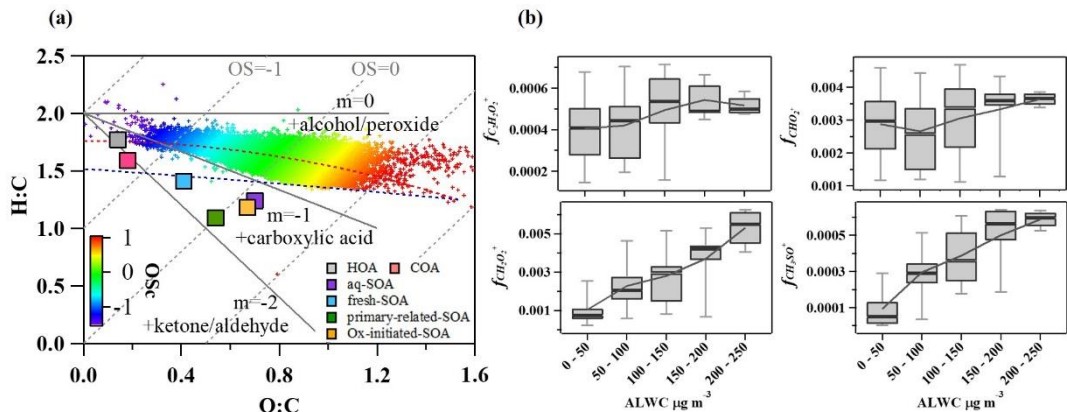


Fig. 8 (a) Van Krevelen diagram for the O:C and H:C ratios of different OA factors (marked with squares) and bulk of OA during summer (marked with plus signs and colored by Osccarbon oxidation state (OSc)); (b) Mass fractions of ion fragments indicative of aqueous-phase processing and oxygenated functional groups (alcohols, carboxylic acids) as a function of ALWC.