# Peer review of "Multiple pathways for the formation of secondary organic aerosol in North China Plain in summer"

_Atmospheric Chemistry and Physics, 2022_

## Author Comment (AC1)

**Response:**

Dear Editor,

We would like to thank you and the reviewers for the time and consideration of our manuscript. We have revised the manuscript following comments raised by the editor and the reviewers and the point-by-point response is attached. Please find our point-by-point response below.

We believe that the manuscript has been significantly improved and adequately addresses the reviewers' comments,

Sincerely yours,

Yifang Gu

On behalf of the authors

**Comment on acp-2022-573**

Anonymous Referee #1

Referee comment on "Multiple pathways for the formation of secondary organic aerosol in North China Plain in summer" by Yifang Gu et al., Atmos. Chem. Phys. Discuss., https://doi.org/10.5194/acp-2022-573-RC1, 2022

General comments:

The paper titled "Multiple pathways for the formation of secondary organic aerosol in North China Plain in summer" by Gu et al. reported a summer field observation in Handan City, in which four types of SOA were resolved using the PMF method dealing with the OA mass spectra data from a a soot particle long time-of-flight aerosol mass spectrometer. In addition, the variations and evolution processes of so-called SOA factors were discussed and their formation pathway were then deduced. Although a series of similar studies using AMS data had been reported in NCP previously, this study provided a more detail source appointment results in distinguish SOA as freshly (less-oxidized) and aged (more-oxidized) factors, and further directly associated with the formation pathway (photochemistry and aqueous-phase). With this, a better understanding of ambient SOA formation and aging in complex pollution with high oxidation capacity were gained. The manuscript was well written and presented clearly. Therefore I recommend the publication of Gu et al. work after some issues were clarified and revised.

**Response:** We thank the reviewer for the positive comments.

Specific and technical comments:

1.       Line 167-170. The new named four SOA factors are interesting, that different from previous studies using same AMS data set resolved by PMF. But the identification of those SOA factors were missed here, or a brief description but not enough appeared in the end of section 3.1. I think the identification of SOA factor is the most important and the foundation of this study, thus a detail description should be provided here. For example, why do the authors directly name the OOA factor as photochemistry and aqueous-phase formed? What the difference between the primary-related SOA and CCOA, both of which showed pronounced peaks of PAH ions?

**Response:** We thank the referee's valuable suggestion and we agree with the referee that the identification is the most important for the description of SOA. In our results, although all of the SOA factors were characterized by higher $m/z$ 44 ($CO_2^+$) and $m/z$ 28 ($CO^+$) signals compared with POA factors, their mass spectrum and temporal trends were noticeably distinguishable, corresponding to different formation mechanisms. In general, the first factor is attributed to aqueous-phase chemistry (aq-SOA) and the other to photo-oxidation chemistry ($O_x$-initiated-SOA, what was named as "phochem-SOA" before, and now has been changed to "$O_x$-initiated-SOA" as suggested). The reason why we directly named it as aq-SOA is that it increased with ALWC but decreased with $O_x$ (Fig. S3), which might be produced/influenced by aqueous-phase chemistry. It was also tightly correlated with $CH_2O_2^+$ ($R^2 = 0.81$) at $m/z$ 46 and $CH_3SO$ ($R^2 = 0.78$) at $m/z$ 63 (Fig. S10), the typical fragment characteristics of ions of aqueous-phase processing products (Tan et al., 2009; Sun et al., 2016; Duan et al., 2022), indicating its characteristics of aqueous phase processing. The aq-SOA exhibits the highest O:C ratios of all factors (0.7) and a higher $f_{CO2+}$ to the total signal of 21.7%, but a low H:C ratio of 1.24 (Fig. 1). In comparison, the $O_x$-initiated-SOA presented an opposite trend with significant increase as a function of $O_x$ but decrease as a function of ALWC (Fig. S3) which is defined as $O_x$-initiated-SOA (influenced by photooxidation chemistry). The pronounced PAH ions have been observed in the primary-related-SOA factor in our study, which was also observed in the POA mass spectrum, such as CCOA and BBOA. While the difference between the primary-related-SOA and CCOA is the oxidation state and mass spectra, the O:C ratio of CCOA is around 0.17 in Li et al., (2017) and its mass spectra can be validated by the high signals at PAH-related m/z such as m/z 77, 91 and 115 (Dall'Osto et al., 2013; Hu et al., 2013) with lower $m/z$ 44 ($CO_2^+$) signal. However, the most abundant peaks in the mass spectra of the primary-related-SOA are at $m/z$ 44, also it is characterized by both lower H:C (1.09) and O:C (0.54) ratios with $CO_2^+$ comprising 14.3%, which are higher than CCOA factor, indicating a typical nature of less oxidized SOA. In addition, their time series and diurnal cycle also show a remarkable difference, which we discussed in Section 3.1 and 3.4.

In the revised manuscript section 3.1, page 6 lines 182-184, we have added the sentence:

"SOA factors were resolved depending on the oxidation state, which correspond to aged SOA and fresh SOA respectively (Jimenez et al., 2009). One factor is attributed to aqueous-phase chemistry (aq-SOA) and the other to photochemistry ($O_x$-initiated-SOA)……"

And in page 7 lines 195-223, we have added a detailed description of the identification of SOA factor, and the manuscript has been modified as follows:

"……The $O_x$-initiated-SOA presented an opposite trend with significant increase as a function of $O_x$ but decreased as a function of ALWC (Fig. S3) which is defined as $O_x$-initiated-SOA (influenced by

photochemistry). As $O_x$ has been shown to be a conserved tracer to represent photo-oxidation chemistry (Xu et al., 2017), the relationship between $O_x$ and $O_x$-initiated-SOA can offer insight into the formation mechanism of SOA associated with the progression of atmospheric photochemical aging (Herndon et al., 2008).

The fresh-SOA showed an increase substantially as ALWC increasing, similar to aq-SOA. Whereas it also showed a slight increase trend following $O_x$ when $O_x < 100$ ppb (Fig. S3). Therefore, both aqueous-phase chemistry and photochemical processing were thought to have positive impacts synchronously on the formation of fresh-SOA. In this study, the fresh-SOA had the lowest atomic O:C ratio of 0.41 and the highest atomic H:C ratio of 1.41 among the four SOA factors, corresponding with the $f_{CO2+}$ of 8.3%, these characteristics are consistent with the global average O:C ratio of LO-OOA of 0.35 $\pm$0.14, Ng et al., 2010), demonstrating it is more fresh SOA. Although the primary-related-SOA constituted a small fraction and showed little variation during P1~P3 (3%~5%), this SOA source is also of particular interest because of its distinctive fragments with high $m/z$ values in the mass spectrum (Fig. 1d). At $m/z < 120$, the primary-related-SOA had higher intensities for $m/z$ 43 (mainly $C_2H_3O^+$) and $m/z$ 44 (mainly $CO_2^+$) than those in POA, indicating a typical nature of less-oxidized SOA. At $m/z > 120$, PAH-derived fragments are clearly evident in the mass spectrum of the primary-related-SOA, as indicated by PAH-like ions (described in SI 1.2) (Dzepina et al., 2007). Previous AMS studies have observed pronounced peaks of PAH ions in POA mass spectra, such as those in coal combustion organic aerosol (CCOA) and biomass burning organic aerosol (BBOA) (Hu et al., 2016b; Zhao et al., 2019), but rarely in SOA. This observation implies that the factor may be related to the POA originated from domestic coal combustion and here it is termed as primary-related-SOA (Xu et al., 2006). Moreover, this SOA factor exhibited relatively better correlations with some gaseous pollutants (Fig. S4), such as CO ($R = 0.6$) and $NO_2$ ($R = 0.5$), and also tracked with HOA ($R = 0.4$). These observations suggest that the primary-related-SOA might be transformed from locally emitted POA as a non-negligible source to SOA."

2.      Line 260-262. The transformation processes of POA and fresh SOA factor to phochem-SOA is interesting, it deserve more discussions here. Exploring the diurnal pattern of these factors during a special episode would be a choice, such as that conducted in Li et al. 2020. https://doi.org/10.1016/j.atmosenv.2019.117070.

**Response:** We thank the referee's valuable suggestion. As suggested by the referee, we have further analyzed the diurnal pattern of more factors and parameters to give a more specific discussion during different periods to explore the transformation processes.

As shown in Fig. R1 below, firstly we explored the transformation through photochemistry from POA and fresh-SOA to $O_x$-initiated-SOA. During P2 as $O_x$ increased, the mass loadings of $O_x$-initiated-SOA showed a substantially increasing trend when $O_x$ was > 30 ppb. As a comparison, the mass concentrations of other OA factors except $O_x$-initiated-SOA showed decreasing trends as $O_x$ increased, suggesting the potential transformation from POA factors and fresh SOA factors to $O_x$-initiated-SOA. Additionally, the average concentration of $O_x$-initiated-SOA increased continually during daytime with a high average growth rate of 0.8 $\mu$g m$^{-3}$ h$^{-1}$, corresponding to the high $O_x$ concentration. In contrast, the concentrations and the contributions of other SOA factors deceased continuously at the same time (Fig. R1). The opposite trends between $O_x$-initiated-SOA and other OA factors from 7:00 to 15:00 LT

suggest that some part of POA and fresh-SOA may convert to $O_x$-initiated-SOA by photochemical oxidation, as indicated by a small afternoon peak of the more oxidized tracer $CO_2^+$ (*m/z* 44) and decrease in a less oxidized tracer $C_2H_3O^+$ (*m/z* 43) (Fig. R2). As the result, with lower water content in the particles, the high solar radiation and high $O_x$ concentration during daytime associated with a relatively high degree of oxygenation of OA suggest that gas-phase oxidation and partitioning processes are probably the dominating process in SOA formation during P2.

In the revised manuscript page 9 lines 290-297, we have now added a series of discussion through daytime photochemical evolutions in the typical episode with high-$O_x$ period (P2), and the manuscript has been modified as follows:

[revised manuscript text omitted]

3.      Line 277-282. The aqueous-phase chemistry may also contributed to the fresh-SOA, but if so it is unclear what the difference of aqueous-phase in fresh-SOA and aq-SOA?

**Response:** We thank the referee for pointing this out. As we discussed in the manuscript, there are two types of aqueous-phase chemistry, including nocturnal and daytime processing. Fig. R1 showed the diurnal cycle of the SOA factors and illustrated the different types of aqueous-phase chemistry during daytime and nighttime in high-RH conditions (P3).

First, their time series supported the interpretation of different formation mechanisms for aq-SOA and fresh-SOA related to aqueous chemistry throughout the campaign (Fig 2c in the revised manuscript). For example, a remarkable continuous growth of aq-SOA concentration (from ~0.3 $\mu$g m$^{-3}$ to 25.2 $\mu$g m$^{-3}$) and ALWC (from 3.1 $\mu$g m$^{-3}$ to 486.1 $\mu$g m$^{-3}$) occurred on 24$^{th}$-28$^{th}$ August (Fig. 2d in the revised manuscript), consistent with the continuous increase in RH (reaching over 95%). The ALWC and aq-SOA were strongly correlated (*R*=0.7, Fig. S4 in SI) during the whole campaign. The fresh-SOA, however, showed a great increase occurred on 24$^{th}$-28$^{th}$ August as well but not as significant as aq-SOA. Comparatively, during the P2 (high $O_x$ period), the average concentration of fresh-SOA (4.0$\pm$2.3 $\mu$g m$^{-3}$) was higher than the concentration of aq-SOA (1.8$\pm$2.0 $\mu$g m$^{-3}$), which we discussed in SI section 1.2.

Second, the aq-SOA is identified as it increased with ALWC but decreased with $O_x$ (Fig. S3 in SI), which is likely due to that aq-SOA formation was accelerated and limited only by ALWC. The fresh-SOA showed a substantial increase as ALWC increasing, whereas it also showed a slight increase trend following $O_x$ when $O_x$ < 100 ppb (Fig. S3 in SI), suggesting both the aqueous-phase chemistry and the photochemistry (in the aqueous phase) participated to produce fresh-SOA synchronously. And both daytime radiative conditions and certain amounts of ALWC are required for its formation.

Third, their growth rates are quite different, as indicated by the different diurnal cycles especially in P3. The growth rate of aq-SOA is higher than fresh-SOA during the night (Fig.R1). The whole pattern of aq-SOA characterized by a significant decreasing trend at daytime, indicating that the dark aqueous-phase is dominated for aq-SOA formation P3. However, it is worth noting that relatively equal three peaks were found in the diurnal variation of fresh-SOA in P3. The peaks at around 6:00 and 19:00 LT were much lower than those of aq-SOA, and the peak at around 13:00 LT corresponded to the peak in the diurnal cycle of $O_x$ (Fig. R1). That means, both the dark aqueous-phase chemistry and the daytime photochemical aqueous-phase reactions are important in the formation of fresh-SOA. The previous study also observed that photochemical aqueous-phase reactions during daytime might have played significant roles in rapid daytime OOA formation, especially under humid conditions (Kuang et al., 2020).

To be clearly, in the revised manuscript page 11 lines 347-352, we have now deleted the sentence "Fig. 5e shows that the fresh-SOA has similar increasing trend with aq-SOA as ALWC increased, which suggests that aqueous-phase chemistry might have also played an important role in the formation of fresh-SOA…..." and added more clear discussion about the difference of aqueous-phase in fresh-SOA and aq-SOA in page 13 lines 419-434, the sentence now reads:

"The aqueous-phase chemistry related to fresh-SOA is more complicated. For example, Fig. 5e shows that the fresh-SOA has a similar increasing trend with aq-SOA as ALWC increased, however, it also increased slightly as $O_x$ increased (Fig. 4e), hinting that both the aqueous-phase chemistry and the photochemistry (including that in the aqueous phase) participated to produce fresh-SOA simultaneously……Although there is also a smaller peak around 13:00 LT in P3, the whole pattern of aq-SOA is characterized by decreasing trend at daytime. These results suggest that fresh-SOA could be formed through dark nighttime aqueous-phase reactions, which are partially reversible upon the evaporation of aerosol liquid water, and also formed through photochemical aqueous-phase reactions during daytime. Different from aq-SOA, which is highly correlated and limited with ALWC, two types of aqueous-phase chemistry in daytime and nighttime are dominant pathways to the fresh-SOA growth."

4. Line 289-290. It is subjective to conclude that the photochemistry is more efficient in elevating the oxidation degree of OA, as the correlations were analyzed during different periods (P2 and P3), during which other factors like primary emissions and transportation would be different and also effect the O:C ratio of OA.

Response: We thank the referee's comment. In the revised manuscript page 11 lines 358-360, we have removed the sentence "This result illustrates that photochemistry is more efficient in elevating the oxidation degree of OA than is the aqueous-phase chemistry." And the sentence in the revised manuscript page 16 lines 523 "but the effect of photochemistry was stronger in SOA formation" in the conclusion has also been removed.

5. Line 308-310. It is unclear where the photochemistry formation of SOA occurred. Please clarify it and revise this sentence.

Response: We thank the referee's suggestion. In the revised manuscript page 12 lines 391-406, we have now deleted "……This observation is similar to results in a previous study showing that both aqueous-phase and gas-phase photochemical reactions substantially contributed to the formation of OOA (a surrogate of SOA) during the high-RH period. The rapid daytime $O_x$-initiated-SOA formation in our study probably occurred in the aqueous phase driven by photochemical reactions during daytime under humid conditions with high ALWC……", and the sentence now reads "……Photochemical reactions through both aqueous-phase and gas-phase can contribute substantially to the SOA formation in polluted areas of NCP, and during haze days with high RH and ALWC the aqueous-phase photochemical processes played a dominant role in daytime SOA formation (Kuang et al., 2020). The rapid daytime $O_x$-initiated-SOA formation in our study possibly occurred on the particle surface and in the aerosol liquid water (Ervens et al., 2011) under humid conditions with high ALWC but driven by gas-phase direct photolysis and oxidation by photooxidants under high $O_x$ conditions…..."

6. Line 351-353. I do not think the transformation of POA to SOA could be deduced based solely on the negative correlation of each other. In fact, as showed in Fig 7, the negative correlation between phochem-SOA and POA would be expected, as the formation period of phochem-SOA usually occurred during the noontime when the boundary layer was much developed, while the POA usually decreased via horizontal and vertical diffusion. It is also supported by the better correlations in P2, which defined as high-Ox period and also has better diffusion conditions. Please clarify it.

**Response:** As correctly pointed out by the reviewer, indeed, the negative correlation between $O_x$-initiated-SOA (phochem-SOA used before) and POA can be partially attributed to the change of boundary layer, which leads to the diffusion of POA at noontime. To investigate the relationship between $O_x$-initiated-SOA and POA without the effect of PBL during P2, we divide the concentrations of $O_x$-initiated-SOA and POA by CO. The results are shown in the following scatter plot (Fig. R3). The $O_x$-initiated-SOA/CO and POA/CO still have a negative correlation ($R$= -0.5) when the effects of PBL are removed, but lower than before ($R$= -0.8). Additionally, the diurnal cycle without PBL effect (OA/CO) was shown in Fig. R4. The ratio of $O_x$-initiated-SOA relative to CO increased continually during the daytime, as other SOA factors deceased at the same time (Fig. R4), giving evidence that some part of POA and fresh-SOA may convert to $O_x$-initiated-SOA by photochemical oxidation at noontime.

Actually, despite the correlation, the transformation from POA to $O_x$-initiated-SOA could be supported by the successive oxidation sequence from primary-related-SOA to $O_x$-initiated-SOA, eventually to aq-SOA(discussed in section 3.4 later and Fig. 8 in the revised manuscript). It could also be indicated by a small afternoon peak of the more oxidized tracer $CO_2^+$ (*m/z* 44) and a decrease in a less oxidized tracer $C_2H_3O^+$ (*m/z* 43) during P2 we discussed in section 3.2 and Fig. 7b.

Thus, some of POA might transform to $O_x$-initiated-SOA during photochemical processing can still be inferred, but we agreed that the changes in boundary layers contributed in large extent to the negative correlation between $O_x$-initiated-SOA and POA related components. To be more accurate in this section, in the revised manuscript page 14 lines 448-456, we deleted the sentences:

"……In addition, compared with P1 and P3, a more positive promotion on the $O_x$-initiated-SOA formation was observed in P2 when $O_x$ was more than 40 ppb. These observations confirm the results in section 3.2 that intensive formation of $O_x$-initiated-SOA was not only produced by photochemical oxidation from VOCs at high-$O_x$ levels, but also potentially through the transformation of POA-related components into $O_x$-initiated-SOA…..."

And more explanation has been added and the sentence now reads as follows:

"……However, the better diffusion conditions in P2 might also attribute a great extent to the negative correlation, as the formation period of $O_x$-initiated-SOA usually occurred during the noontime when the boundary layer was much developed, while the POA usually decreased via horizontal and vertical diffusion……."

[Figure]

**Fig. R3** Relationship between the ratio of $O_x$-initiated-SOA/CO and the sum of POA and primary-related-SOA / CO in high $O_x$ period (P2) in this campaign.

.

[Figure]

**Fig. R4** Diurnal cycle of ratios of OA factors relative to CO (OA/CO) in reference period (P1), high $O_x$ period (P2) and high RH period (P3) in this campaign.

**Comment on acp-2022-573**

Anonymous Referee #2

Referee comment on "Multiple pathways for the formation of secondary organic aerosol in North China Plain in summer" by Yifang Gu et al., Atmos. Chem. Phys. Discuss., https://doi.org/10.5194/acp-2022-573-RC2, 2023

This manuscript investigates the formation of secondary organic aerosol in the summertime North China Plain (NCP). The authors observed that both photo-oxidation and aqueous-phase chemistry contribute to the formation of SOA in NCP. Results highlight that the SOA formation is related to residential coal combustion, and the SOA formed could be further oxidized rapidly under high-RH conditions. Overall, the manuscript is well-written. The results are interesting. I recommend the manuscript be considered for publication only after my following comments are fully addressed.

**Response:** The authors thank the referee to review our manuscript and particularly for the constructive comments and suggestions that are very helpful in improving the manuscript. All concerns have been carefully addressed. Below is our point-by-point response to each comment. We also have made most of the changes suggested by the referee in the revised manuscript.

1. What were the concentrations of inorganic cations such as Fe, Mg, and Ca? How would it affect the calculation of ALWC?

**Response:** We thank the referee's comment. In principle, aerosol liquid water content (ALWC) was calculated using the ISORROPIA-II model by inputting the inorganic ions, such as $SO_4^{2-}$, $NO_3^-$, $NH_4^+$, $Cl^-$, $Na^+$, $K^+$, $Ca^{2+}$, $Mg^{2+}$, RH and T, which were selected as the determining factors (Fountoukis and Nenes, 2007). In our study, inorganic aerosol composition ($NH_4^+$, $SO_4^{2-}$, $NO_3^-$, $Cl^-$) from SP-LToF-AMS combined with T and RH were input into the ISORROPIA-II model for simulation, while the ions such as $Na^+$, $K^+$, $Ca^{2+}$, $Mg^{2+}$ were not considered because of instrument limitation (lack of long-term online observation).

According to previous studies, these crustal ions constituted relatively small fractions of aerosol, and had relatively weak effects on ALWC accumulation (Fountoukis and Nenes, 2007;Su et al., 2022).

2. What was the pH of the aerosol particles? Did it contribute to the photochemical and aqueous formation of SOA? Please discuss.

**Response:** Thanks for the referee's suggestion. The pH can be estimated from water and $H^+$ concentrations simulated by ISORROPIA (pH=$-\log_{10}$($1000H^+$/water) (Guo et al., 2015). In our study, the pH of aerosol particles is estimated ~2.7 through this method, which is much lower than others' results. According to previous studies (Wang et al., 2020; Zhang et al., 2021), the pH of the aerosol particles can be varied from 4.5-5.2 on the North China Plain (NCP), at which Handan city locates. It is might because of the lack of $NH_3$ gas in our simulations. The results without $NH_3$ have a larger deviation in pH calculations and may underestimate the pH value., thus the effect of pH contribution to the photochemical and aqueous formation of SOA is not shown, which will be considered more in our further campaign.

We further refer to the experimental results to discuss the possible effect of the pH of the aerosol particles on the formation of SOA. As it is stated in Zhou et al., (2019), aerosol acidity may influence the pH-dependent aqueous phase processes that take place in the aerosol condensed phase, which in turn contribute to the formation and growth of SOA. We also infer that hydroxyl radical oxidation (OH) can play an equal or more important role compared to triplet excited states of organic compounds (3C), since 3C reaction is only active when pH < 4 (Smith et al., 2013). Moreover, SOA yields due to organic

photochemistry in the aerosol aqueous phase from Acetylene ($C_2H_2$) can be inhibited by the possible existing sulfate acid, which can lead to low pH (1.0-3.0) (Volkamer et al., 2009). The previous modeling study (McNeill et al., 2012) also found that pH can show different effects to aqueous-phase secondary organic aerosol and organosulfate formation in atmospheric aerosols: Formation of SOA in the aqueous phase increase with the decreasing of the pH values under the low-$NO_x$ condition, while it can be independent to the pH values with the high-$NO_x$ condition.

Overall, the contribution of the pH of the aerosol particles to the formation of SOA is a complicated but important and interesting research topic, which we will focus on in our future work.

3. Lines 167-170: A sentence is needed here to briefly illustrate how these SOA factors were defined. For example, what was fresh SOA? Was it LOOOA?

**Response:** We thank the referee's suggestion. As response above and suggested by referees, in the revised manuscript section 3.1, page 6 lines 182-185, and in page 7 lines 196-223, the manuscript has been modified. The identification and description of these four SOA factors have been added. For these four SOA factors, their mass spectrum and temporal trends were noticeably distinguishable, corresponding to different formation mechanisms, more discussion on details could also be found in SI section 1.2.

4. It seems that fresh-SOA and aq-SOA were quite similar (Figures 2, 4, 5, S4). Why did the authors separate them into two factors? Please clarify.

**Response:** We thank the referee's comment.

First, the mass spectrums are different in Fig 1 (in the revised manuscript). Although the mass spectra of the aq-SOA and fresh-SOA factors all contain a prominent peak at $m/z$ 44 (mainly $CO_2^+$), representing their SOA features. The mass spectrum of fresh-SOA shows a significantly higher peak at $m/z$ 43 (mainly $C_2H_3O^+$) and the lowest O:C ratio (0.41) in all the SOA factors, suggesting the less oxidized property. While the aq-SOA exhibits the highest O:C ratios of all factors (0.7) and a higher $f_{CO2+}$ to the total signal of 21.7%, suggesting the more oxidized SOA property.

Second, the time series are different. During on 24th-28th August (Fig. 2d in the revised manuscript), both fresh-SOA and aq-SOA concentration showed remarkably continuous increase together with ALWC (from 3.1 to 486.1 $\mu g$ $m^{-3}$, consistent with the continuous increase in RH (reaching over 95%). However, the growth rate of fresh-SOA was much lower aq-SOA. In comparison, during P2 (high $O_x$ period), the average concentration of fresh-SOA (4.0±2.3 $\mu g$ $m^{-3}$) was higher than the concentration of aq-SOA (1.8±2.0 $\mu g$ $m^{-3}$). It could be supported by different formation mechanisms for aq-SOA and fresh-SOA related to aqueous chemistry throughout the campaign (Fig 2c in the revised manuscript).

Third, the formation mechanisms are different. The aq-SOA was identified as it increased with ALWC but decreased with $O_x$ (Fig. S3 in SI), which might be produced in the aqueous-phase events and influenced by aqueous-phase chemistry. The fresh-SOA increases substantially as ALWC increasing, similar to aq-SOA. Whereas it also showed a slight increase trend following $O_x$ when $O_x$ <

100 ppb (Fig. S3 in SI). Therefore, ALWC and the oxidants are critical for fresh-SOA formation and both the aqueous-phase chemistry and the photochemistry (including that in the aqueous phase) participated to produce fresh-SOA synchronously. Moreover, aq-SOA shows a quite clear peak during nighttime (Fig. R1), indicating that the dark aqueous-phase is dominated for aq-SOA formation under high-RH conditions. However, it is worth noting that three peaks were found in the diurnal variation of fresh-SOA in P3. The peaks at around 6:00 and 19:00 LT were similar to those of aq-SOA, while the peak at around 13:00 LT corresponded to the peak in the diurnal cycle of $O_x$ (Fig. R1). Although there is also a smaller peak around 13:00 LT in P3. The whole pattern of aq-SOA is characterized by decreasing trend at daytime. This three-peak diurnal pattern suggests it can be formed through dark aqueous-phase chemistry during nighttime and photochemical aqueous-phase reactions during daytime, these two types of aqueous-phase chemistry are both important in the formation of fresh-SOA.

The discussions about the difference have been added and modified in the revised manuscript on page 13 lines 419-434 as follows:

"The aqueous-phase chemistry related to fresh-SOA is more complicated, requiring both daytime radiative conditions and certain amounts of ALWC in nighttime. For example, Fig. 5e shows that the fresh-SOA has a similar increasing trend with aq-SOA as ALWC increased, however, it also increased slightly as $O_x$ increased (Fig. 4e), hinting that both ALWC and the oxidants are critical for fresh-SOA formation and both the aqueous-phase chemistry and the photochemistry (including that in the aqueous phase) participated to produce fresh-SOA simultaneously. It is worth noting that three peaks were found in the diurnal variation of fresh-SOA in P3. The peaks at around 6:00 and 19:00 LT at night were similar to those of aq-SOA and lower than it, while the peak at around 13:00 LT is consistent with the peak in the diurnal cycle of $O_x$ (Fig. 6). Although there is also a smaller peak around 13:00 LT in P3, the whole pattern of aq-SOA characterized by decreasing trend at daytime. These results suggest that fresh-SOA could be formed through dark nighttime aqueous-phase reactions, which are partially reversible upon the evaporation of aerosol liquid water, and also formed through photochemical aqueous-phase reactions during daytime. Different from aq-SOA, which is highly correlated and limited with ALWC, two types of aqueous-phase chemistry in daytime and nighttime are dominant pathways to the fresh-SOA growth."

5. Lines 212-224: The authors mentioned that the primary-related SOA might be transformed from locally emitted POA, as suggested by the PAH ions in the primary-related SOA. What was the correlation of PAH ions in the primary-related SOA and POA? It seems that the patterns are quite different.

In addition, in my view, the correlations between the "primary-related SOA" and CO, NO2, and HOA were not high. The highest R-value was only 0.6, with R2 < 0.5. Could it be just primarily emitted OA?

Also, why did the "primary-related SOA" mostly peak at night? Please explain.

**Response:** We thank the referee's comment.

In our study, polycyclic aromatic hydrocarbons (PAHs) fragments were found at mass-to-charge

ratios (*m/z*) 152, 165, 178, 189, 202, 216, 226 + 228, 240 + 242, 250 + 252, 264 + 266 and 276 + 278 (Dzepina et al., 2007). The signatures of PAH ions pattern in the mass spectrum of primary-related-SOA (Fig. 1 in the revised manuscript) are not correlated very well with it in HOA and COA of our PMF results POA. Usually, pronounced peaks of PAHs ions were observed in POA spectra such as CCOA (coal combustion) and BBOA (biomass burning) in previous AMS studies (Hu et al., 2016; Zhao et al., 2019). However, the factors of CCOA and BBOA don't exist in our study. It is perhaps due to: (1) In summer of NCP regions, there are not many activities like biomass burning and coal combustion, resulting in very low concentration of PAH ions concentration; (2) Even there are combustion activities in the sampling site, the relatively high atmosphere oxidation capacity could facilitate the rapid oxidized processing from originated POA to SOA, and the oxidation of PAHs being involved in the conversion of POA to primary-related-SOA and aq-SOA. This transformation process will lead to the different patterns of PAHs in POA and SOA, which was supported by Budisulistiorini et al., (2021), who found that POA can be oxidized by multiphase reactions forming OPOA and the degradation of high molecular weight (HMW) species from the oxidation process.

We agree that *R*=0.6 is not a quite high correlation in the atmosphere, the correlation between primary-related-SOA and other species here, could just give an assumption first that the original primary-related-SOA was partly related by some primarily emitted precursors, such as CO, $SO_2$, $NO_2$ and HOA. This correlation could not directly indicate if it is just originated from primary-related-SOA emitted. Similar results were obtained by Rivellini et al., (2020), which showed the oxygenated-HOA (O-HOA) or oxygenated-CCOA reported could be found through co-emitted with HOA and/or produced rapidly via oxidation of POA near emission sources by oxygenated part of combustion particles. However, O-HOA still showed distinctive time series and oxygenation levels compared with HOA (r of 0.17 and O/C of 0.07). Actually, in our study, the major pathway of this primary-related-SOA formation might be originally associated with primary emission or the fast oxidation/transformation of fresh POA precursors. This similar processing was found in laboratory experiments, such as the results in Zhang et al., (2021) showed that some SOA factors were defined as "urban-lifestyle SOAs" because they could be derived from some POA exhaust such as vehicles and cooking.

More evidence about its primary features was found in our discussion. Its pollution pattern was similar as some primary precursors such as CO, $SO_2$ and $NO_2$, where higher concentrations appeared with the weak west wind (Fig. S5 in SI), suggesting that primary-related-SOA was a regional source and partly related to the emissions transported from the industrial zone.

Also, considering primary-related-SOA is a regional source consistent with CO, $SO_2$ and HOA, originated from the western area, the peak during night is probably due to the lower boundary layer at night. The decreased PBL will contribute to the accumulation of both primary and secondary pollutants. As shown in Fig. R4, without the boundary layer effects, less variation was found in primary-related-SOA's diurnal pattern, especially at night, supporting the regional source of it.

6. Lines 274-277: If SP-LTof-AMS collected PM2.5, how can aq-SOA in droplets be sampled and analyzed? Please provide the size distribution of the particle sampled. The authors can compare the size distribution of aq-SOA to sulfate or other inorganic ions to support their conclusions.

**Response:** We thank for the referee's suggestion. In our campaign, aerosol particles below 2.5 $\mu$m were dried before sampling into the AMS, and the aq-SOA inside the droplet here was also measured by dry particles and then analyzed in AMS. Here, we further provide the size distribution of sulfate and the typical tracer ion of aq-SOA, *m/z* 44 (shown in the Fig. R5). Considering we couldn't obtain the high resolution of PTOF data from our instrument, the size distribution of aq-SOA cannot be provided currently. However, the size distribution of *m/z* 44 (Fig. R5) was plotted to represent aq-SOA because it is the most pronounced fragment in the mass spectrum of aq-SOA and it is characteristic of ions for SOA generally.

As shown in Fig. R5, the size distribution of aq-SOA (600~700 nm) is quite similar to the size distribution of sulfate, which ensures that the sampled particles are successfully dried and the chemical compositions of most fog droplets (typically below 2.5 $\mu$m) can be detected by the AMS (Ervens et al.,2011; Shen et al., 2018). The similar size distributions of aq-SOA with sulfate were consistent with the results in Wang et al., (2021). Previous evidence indicates that sulfate was produced rapidly by cycling of air parcels through fog and cloud (Wang et al., 2020), and the same could apply to aq-SOA resulting in a shift to larger particle sizes. The size distribution of PMF-OA factors is developed now in our group and we will provide a more comprehensive analysis in the further.

[Figure]

**Fig. R5** Average size distributions of sulfate ($SO_4^{2+}$), *m/z* 44 from AMS.

7. Figure S3: Please explain why aq-SOA concentration decreased with ozone concentration, and why Ox-initiated-SOA concentration decreased with ALWC?

**Response:** We thank the referee's comment.

In our study, the formations of aq-SOA and $O_x$-initiated-SOA are mainly related to the aqueous-phase processing and photochemical production, respectively. All these characteristics illustrate that the high

ALWC condition promoted the formation of aq-SOA but ALWC suppressed $O_x$-initiated-SOA formation. Also high $O_x$ condition promoted the $O_x$-initiated-SOA formation but $O_x$ suppressed aq-SOA formation. This conclusion is further supported by the correlation in aq-SOA/$O_x$-initiated-SOA with ALWC/$O_x$ (Fig. S4 in SI). The aq-SOA correlates well with ALWC, whereas a poor correlation was observed between aq-SOA and $O_x$. Consistently, The $O_x$-initiated-SOA showed a negative correlation with ALWC but correlates well with $O_x$. As it is shown in Fig. R1, the trends of diurnal patterns of RH (related to ALWC) and $O_x$ are opposite. Therefore, the high $O_x$ condition did not promote the formation of aq-SOA and the mass decrease is likely due to the concentration decrease in $O_x$, further supporting the dominant role of aqueous-phase processing in aq-SOA production. And the concentration of $O_x$-initiated-SOA decreased when ALWC is high because of the low concentration of $O_x$, suggesting the important role of photochemical processing in $O_x$-initiated-SOA production.

8. Figure 5: How could aq-SOA be formed when ALWC was 0?

**Response:** We thank the referee's comment. Actually, Figure 5 in the revised manuscript represents the variations in the mass fractions and mass concentrations of different OA factors as functions of ALWC. The data were binned according to the ALWC concentration (20 $\mu$g m$^{-3}$ increment in P1 and P2, 50 $\mu$g m$^{-3}$ increment in P3). Specifically, a series of intervals for ALWC, e.g., [0,20], [20,40]......were set. Then we calculated the average OA concentrations for each interval and depict the values at the start point of the interval. Therefore, the aq-SOA value at ALWC=0 represents the averaged concentration of aq-SOA when ALWC is ranged from 0 to 20 or 50. Furthermore, although the formation of aq-SOA is highly correlated with ALWC, it can still be generated even the ALWC is at low values. In the revised manuscript, we have added more details about the figures and will make it clear in the revision.

The figure caption of Figure 4 now reads as follows:

"**Fig. 4**…….The data were binned according to $O_x$ concentration (10 ppb increment in P1, 20 ppb increment in P2 and P3)."

The figure caption of Figure 5 now reads as follows:

"**Fig. 5**…….The data were binned according to the ALWC concentration (20 $\mu$gm$^{-3}$ increment in P1 and P2, 50 $\mu$gm$^{-3}$ increment in P3)."

9. Figure 6: There was a substantial amount of phochem-SOA at night. Can the authors discuss if it was possible that "phochem-SOA" can also be formed at night? Could it be just SOA formed via ozone oxidation? Maybe using a different name for "phochem-SOA" would avoid confusion.

**Response:** We thank the referee's suggestion.

Usually, the photochemical oxidation occurred in daytime, corresponding with the afternoon peak in diurnal pattern of $O_x$ and $O_3$ (Fig. R8). In our study "phochem-SOA" increased rapidly during daytime with a high average growth rate of 0.8 $\mu$g m$^{-3}$ h$^{-1}$ caused by high $O_x$ concentration, high temperature and strong solar radiation synergistically. However, it shows a decreasing trend during

nighttime (Fig. R1), even without the BPL effect (Fig. R4). The substantial amount of "phochem-SOA" during nighttime is probably due to the accumulation of SOA formation at daytime.

The relationship between OOA and Odd oxygen ($O_x = O_3 + NO_2$) can be used as a metric to characterize SOA formation mechanisms associated with ozone production chemistry (Xu et al., 2017). The pervious observation found that a strong correlation between SOA and $O_x$ was linked to the extent of photochemical oxidation in an air mass (Kuang et al., 2020). From our results, "phochem-SOA" correlated tightly with $O_x$, suggesting its formation is driven by photochemical processing characteristics with strong solar radiation when it forms. The name of this factor has been changed to "$O_x$-initiated-SOA" to avoid confusion as suggestion. Actually, we plan to conduct the photolysis as well as nocturnal chemistry effects in a laboratory in the future to further investigate the SOA formation.

10. Most of the results provided in this study are based on correlation analysis. Although I believe that correlations provide valuable insights, the results from correlations solely may not be conclusive and convincing enough. Can the authors provide additional evidence (e.g., from the perspective of chemical composition and tracers) to support the conclusions of this study?

**Response:** We thank the referee's comment.

We agree with the referee that correlation analysis did make a great of contribution to the results in our study. However, other elucidations and characterizations have also been provided as evidence based on the comparison of the chemical compositions and meteorological parameters in different pollution periods, the diurnal variations, the growth rate of SOA factors, and the evolution of OA oxidation degree from Van Krevelen plot.

In particular: (1) The comparison of SOA factors and meteorological parameters suggested that during high-$O_x$ period (P2), enhanced SOA formation was promoted by photochemistry, making the Ox-initiated-SOA the major source of SOA during P2. During the high-RH period (P3), rapid SOA production through the aqueous-phase chemistry was observed, leading to the increase in the degree of oxygenation in total OA; (2) From the diurnal cycle, the efficient SOA formation from photochemistry ($O_x$-initiated-SOA) dominated the daytime (65% to OA) as the concentration of $O_x$-initiated-SOA increased from morning to afternoon with an average growth rate of 0.8 $\mu$g m$^{-3}$ h$^{-1}$. Moreover, strong nocturnal aqueous-phase SOA formation (aq-SOA) played a significant role in SOA production (45% to OA) with a nighttime growth rate of 0.6 $\mu$g m$^{-3}$ h$^{-1}$. Meanwhile, an equally fast growth rate of 0.6 $\mu$g m$^{-3}$ h$^{-1}$ of $O_x$-initiated-SOA from daytime aqueous-phase photochemistry was also observed, which contributed 39% to OA, showing that photochemistry in the aqueous phase is also a non-negligible pathway in summer; (3) The ratios of O:C and H:C both increase in the succession from primary-related-SOA to $O_x$-initiated-SOA and eventually to aq-SOA, supporting a successive oxidation sequence from primary-related-SOA to aq-SOA. The PAH-like ions found in the mass spectrum suggested transformation via hydroxylation of the aromatic ring or ring-breaking oxidation of aromatic POA species through aqueous-phase chemistry.

In addition, to further support the conclusions of this study, a more detailed analysis of the initial pollution stage and multiple formation pathways are elucidated in the revised manuscript as suggested by the referees, including the diurnal pattern of more factors and parameters to give a more specific

discussion during different periods (Fig. R1). Meanwhile, the related fragment ions in high-resolution organic mass spectra have been further analyzed (Fig. R6 and Fig. R7) to elucidate the evolution and transformation between different types of SOA.

In the revised manuscript page 15 lines 499-513, we have now added the diurnal cycle of fragment ions and mass fractions (Fig. R4) in ion fragments indicative of oxygenated functional groups (carboxylic acids and glyoxal) as a function of ALWC in both P2 and P3 (Fig. R6b), and the manuscript has been modified as follows:

"Specifically, the organic fragments and mass spectrum evolution of OA were analyzed to illuminate the transformation in photochemical processing and aqueous-phase chemistry. Fig. 8 shows the mass fractions of $CH_2O_2^+$, $CH_3SO^+$, $HCO_2^+$, and $C_2H_2O_2^+$ ion fragments in OA as a function of ALWC. The aq-SOA was tightly correlated with $CH_2O_2^+$ ($R^2 = 0.81$) at $m/z$ 46 and $CH_3SO$ ($R^2 = 0.78$) at $m/z$ 63 (Fig. S10), Consistently, both of them showed increase trends as ALWC increasing, similar as aq-SOA, which indicating typical fragment characteristics of ions of aqueous-phase processing products (Tan et al., 2009; Sun et al., 2016; Duan et al., 2021). The intensities of $HCO_2^+$ ($m/z$ 45), a common fragment ion of carboxylic acids, is associated with aqueous oxidation of aromatic compounds. $C_2H_2O_2^+$ ($m/z$ 58) is a tracer ion for glyoxal, which could be a ring-breaking product from the aqueous-phase oxidation of PAHs. The increasing trends of these ions with ALWC suggest that water-soluble organic species such as carboxylic acids and glyoxal are produced as components of aq-SOA following aromatic oxidation and ring breaking. Moreover, the concentration of PAHs increased with the increase of ALWC (Fig. S11), consistent with the oxidation of PAHs from ring-breaking reactions that can take place in the aqueous phase and being involved in the conversion to aq-SOA."

Moreover, the transformation from POA or fresh SOA to aged SOA could be indicated by the increase of the more oxidized tracer $CO_2^+$ ($m/z$ 44) and the decrease in a less oxidized tracer $C_2H_3O^+$ ($m/z$ 43) during daytime of P2 and nighttime of P3. In the revised manuscript page 9 lines 290-297, and page 10 lines 308-325, we have now added a series of discussion through daytime photochemical evolutions in the typical episode with high-$O_x$ period (P2), and the manuscript has been modified as follows:

[revised manuscript text omitted]

Additionally, Fig. 8 in the revised manuscript is now updated as Fig. R6 below, and Fig. S11 in the revised supplementary information is now updated as Fig. R7 below.

[Figure]

**Fig. R6** (a) Van Krevelen diagrams for the O:C and H:C ratios of different OA factors and bulk of OA during summer (markers with diamonds; (b) Mass fractions of ion fragments indicative of aqueous-phase processing and oxygenated functional groups (alcohols, carboxylic acids) as a function of ALWC.

[Figure]

**Fig. R7** Mass concentration of PAHs ions as a function of ALWC.

---

## Editor Decision (ED1)

**Editor minor revisions request**

The authors have successfully addressed the referees' questions and brought relevant contribution into SOA formation processes in North China Plain. For acceptance, I kindly ask the authors to clarify the points below.

1) Referee # 2, question 1: Could the authors, please, add to the manuscript the information that those cations were not measured and justify that statement by using the references from previous studies that were mentioned in the reply?

2) Referee # 2, question 8: I think the reviewer has a point in here, which is, in Fig 5, it is possible to see a consistent mass concentration of aq-SOA even when ALWC is very low. Could it mean that those fractions (~10-20%) at low ALWC indicate the presence of a background SOA, not necessarily dependent on the ALWC, or …? I recommend that some brief explanation could be added to the manuscript.

3) It is clear that the Ox-initiated-SOA is related to photochemistry, however, could the authors, please, help the reader to properly understand what this terminology actually stands for? For example, if it corresponds to SOA formation initiated due to the presence of Ox, or …?

---

## Author Response (AR2)

Editor minor revisions request
The authors have successfully addressed the referees' questions and brought relevant contribution into SOA formation processes in North China Plain. For acceptance, I kindly ask the authors to clarify the points below.

1) Referee # 2, question 1: Could the authors, please, add to the manuscript the information that those cations were not measured and justify that statement by using the references from previous studies that were mentioned in the reply?

We thank for the suggestion. The statement and references have both been added in the revised manuscript page 6 line 162-165.

2) Referee # 2, question 8: I think the reviewer has a point in here, which is, in Fig 5, it is possible to see a consistent mass concentration of aq-SOA even when ALWC is very low. Could it mean that those fractions (~10-20%) at low ALWC indicate the presence of a background SOA, not necessarily dependent on the ALWC, or …? I recommend that some brief explanation could be added to the manuscript.

We thank for the suggestion. Actually, this consistent mass concentration of aq-SOA is just because the intervals used. The data were binned according to the ALWC concentration (14 $\mu$g m$^{-3}$, 20 $\mu$g m$^{-3}$, 40 $\mu$g m$^{-3}$ increment in P1, P2 and P3), but over 80% of ALWC data were placed in the first interval (ranged from 0 ~ 40 $\mu$g m$^{-3}$) because ALWC showed low mass loading in most period time (Fig.2). That makes the first average aq-SOA concentration was not too low as expected, and it seems like a consistent mass concentration of aq-SOA when ALWC is very low. If we use narrower binned interval we could see the difference. For example, in Fig.S1, we used the narrower binned interval of 5 $\mu$g m$^{-3}$ to calculate, the average aq-SOA at the first interval was low enough, and it showed increased trend as the function of ALWC as well. It indicates that aq-SOA are highly dependent on ALWC, and it can still be generated even the ALWC is at low value. Even if there is a background SOA, it will be smaller fraction.

However, we can't replace the Fig. 5 with this narrower bin interval on account of the growth of ALWC is too rapid during the campaign, which will cause the loss of many intervals and calculated results if we use narrower bin intervals.

To be more clear, some explanations have been added to the revised manuscripts page 11 line 343-349 as below.
"Note that there are still consistent mass concentrations of aq-SOA even when ALWC is very low (data interval ranging from 0~40 $\mu$g m$^{-3}$), which is due to that over 80% of ALWC mass concentration were loaded in the first interval, leading to a higher mean value of aq-SOA mass concentration. Actually ALWC showed quite low mass loading in most period time but increased dramatically during P3, yet the time series of aq-SOA and ALWC were remarkably well correlated throughout the entire campaign (*R*=0.7, Fig. S4) rather than a strong correlation observed only in P3."

[Figure]

Fig.S1 The mass concentration of aq-SOA ($\mu g\ m^{-3}$) as functions of ALWC ($\mu g\ m^{-3}$) .

3) It is clear that the Ox-initiated-SOA is related to photochemistry, however, could the authors, please, help the reader to properly understand what this terminology actually stands for? For example, if it corresponds to SOA formation initiated due to the presence of Ox, or …?

We thank for the suggestion. More interpretation has been added in the revised manuscript page 7 line 199-205, and now it reads as follows:

"The $O_x$-initiated-SOA in our study is photochemical production SOA whose formation initiated with the presence of $O_x$. As $O_x$ has been shown to be a conserved tracer to during photochemical processing (Xu et al., 2017), the relationship between $O_x$ and $O_x$-initiated-SOA can represent a metric to characterize SOA formation mechanisms associated with ozone production chemistry SOA (Herndon et al., 2008). $O_x$-initiated-SOA presented an opposite trend with significant increase as function of $O_x$ but decreased as a function of ALWC (Fig. S3), suggesting the dominant role of photochemical processing in the formation of $O_x$-initiated-SOA."